# Isoform-specific subcellular localization and function of protein kinase A identified by mosaic imaging of mouse brain

Ronit Ilouz[1]*, Varda Lev-Ram[1], Eric A Bushong[2], Travis L Stiles[3], Dinorah Friedmann-Morvinski[4,5], Christopher Douglas[3], Jeffrey L Goldberg[3], Mark H Ellisman[2,6], Susan S Taylor[1,7]*

[1]Department of Pharmacology, University of California, San Diego, La Jolla, United States; [2]Center for Research in Biological Systems, National Center for Microscopy and Imaging Research, University of California, San Diego, San Diego, United States; [3]Department of Ophthalmology, Shiley Eye Center, University of California, San Diego, La Jolla, United States; [4]Laboratory of Genetics, The Salk Institute for Biological Studies, La Jolla, United States; [5]Department of Biochemistry and Molecular Biology, George S Wise Faculty of Life Sciences, Tel Aviv University, Tel Aviv, Israel; [6]Department of Neurosciences, University of California, San Diego School of Medicine, La Jolla, United States; [7]Department of Chemistry and Biochemistry, University of California, San Diego, La Jolla, United States

*For correspondence: rilouz@ucsd.edu (RI); staylor@ucsd.edu (SST)

**Competing interests:** The authors declare that no competing interests exist.

**Abstract** Protein kinase A (PKA) plays critical roles in neuronal function that are mediated by different regulatory (R) subunits. Deficiency in either the RI$\beta$ or the RII$\beta$ subunit results in distinct neuronal phenotypes. Although RI$\beta$ contributes to synaptic plasticity, it is the least studied isoform. Using isoform-specific antibodies, we generated high-resolution large-scale immunohistochemical mosaic images of mouse brain that provided global views of several brain regions, including the hippocampus and cerebellum. The isoforms concentrate in discrete brain regions, and we were able to zoom-in to show distinct patterns of subcellular localization. RI$\beta$ is enriched in dendrites and co-localizes with MAP2, whereas RII$\beta$ is concentrated in axons. Using correlated light and electron microscopy, we confirmed the mitochondrial and nuclear localization of RI$\beta$ in cultured neurons. To show the functional significance of nuclear localization, we demonstrated that downregulation of RI$\beta$, but not of RII$\beta$, decreased CREB phosphorylation. Our study reveals how PKA isoform specificity is defined by precise localization.

## Introduction

Precise spatiotemporal regulation of signaling molecules is central to the intricacies of signal transduction. cAMP-dependent protein kinase (PKA), which is ubiquitously expressed in every mammalian cell, regulates numerous signaling pathways and is critical for many neuronal functions. These include learning and memory (*Kandel, 2012*) and multiple forms of synaptic plasticity (*Abel et al., 1997*; *Yasuda et al., 2003*). Pharmacological or genetic inhibition of PKA severely affects the induction of hippocampal long-term potentiation and inhibits synaptic plasticity and long-lasting memory (*Abel et al., 1997*). A reduction in PKA signaling contributes to the etiology of several neurodegenerative diseases, including Alzheimer's disease and Parkinson's disease (*Dagda and Das Banerjee, 2015*; *Howells et al., 2000*). Phosphorylation mediated by cAMP signaling is critically involved in

cell growth, differentiation, apoptosis, synaptic release of neurotransmitters and gene expression (*Skalhegg and Tasken, 2000*), and a large part of the functional diversity of this kinase results from isoform diversity. Eukaryotic cells express multiple forms of PKA regulatory (R) and catalytic (C) subunits. PKA holoenzyme consists of an R-subunit dimer bound to two C- subunits ($R_2C_2$). The biochemical and functional features of PKA holoenzymes are largely determined by the structure and the biochemical properties of the regulatory subunits (*Ilouz et al., 2012*; *Taylor et al., 2012*; *Zhang et al., 2012*).

There are two classes of R-subunits, RI and RII, which are classified into $\alpha$ and $\beta$ subtypes (*Døskeland et al., 1993*; *McKnight et al., 1988*). Each isoform is encoded by a unique gene and preferentially expressed in different cells and tissues. RI$\alpha$ and RII$\alpha$ are ubiquitously expressed in every cell, whereas RI$\beta$ and RII$\beta$ expression is more tissue restricted (*Cadd and McKnight, 1989*). RI$\beta$ is expressed in brain and spinal cord (*Cadd and McKnight, 1989*). RII$\beta$ is predominantly expressed in brain, endocrine, fat, liver and reproductive tissues (*Cadd and McKnight, 1989*; *Jahnsen et al., 1986*). The four R-subunits are functionally non-redundant. Depletion of either the RI$\beta$ or the RII$\beta$ gene in mice resulted in specific neuronal defects. RII$\beta$ knockout mice display defects in motor behavior and loss of PKA-mediated neuronal gene expression (*Brandon et al., 1998*). Hippocampal slices from RI$\beta$ null mice show a severe deficit in long-term depression and depotentiation at the Schaffer collateral–CA1 synapse. Despite a compensatory increase in the RI$\alpha$ protein levels, hippocampal function was not rescued (*Brandon et al., 1995*).

The composition and specific structural and biochemical properties of PKA holoenzymes ($R_2C_2$) account, in part, for differential cellular responses to discrete extracellular signals that activate adenylate cyclase (*Taylor et al., 2012*). Space-restricted kinase activation provides an extra layer of specificity in PKA signaling. PKA is typically targeted at specific intracellular microdomains through interactions with A-Kinase Anchoring Proteins (AKAPs). Many AKAPs have been identified together with their specific requirements for selective binding to regulatory subunits (*Sarma et al., 2010*; *Wong and Scott, 2004*). This spatio-temporal regulation determines the access of proteins to interacting binding partners. AKAPs provide a control mechanism that directs, integrates and locally attenuates the cAMP-initiated cascade. The hallmark signature motif of the AKAPs is an amphipathic helix that binds tightly to the dimerization and docking (D/D) domain of the R-subunits. Recently, a point mutation in the D/D domain of the RI$\beta$ gene has been associated with a new neurodegenerative disease that presents with dementia and Parkinsonism, characterized by specific and abundant accumulation of RI$\beta$ in neuronal inclusions (*Wong et al., 2014*).

Rigorous cellular characterization of protein localization is a necessary step if we aim to understand PKA function in a physiological context. To date, relatively few efforts have attempted to define the subcellular localization of endogenous proteins systematically using imaging-based techniques. Currently, RI$\beta$, the isoform that has a unique role in synaptic plasticity (*Brandon et al., 1995*) and has been associated with a neurodegenerative disease, is the least-studied PKA isoform. RI$\beta$ spatial localization has not been systematically studied due to antibody cross-reactivity with RI$\alpha$. Furthermore, most of the available RII$\beta$ localization data are focused on specific regions of interest; thus the global context of the protein localization is lost.

Subcellular cAMP signaling domains are defined by the distinct environments within cellular organelles. The dogma of cAMP-PKA signaling in the nucleus states that, upon cAMP-induced activation of the cytosolic PKA holoenzyme, the C-subunit dissociates from the R-subunit in an isoform-specific manner and translocates into the nucleus by diffusion (*Harootunian et al., 1993*). Contradictory, reports have increasingly proposed the existence of resident pools of nuclear PKA holoenzyme (*Jarnaess et al., 2009*; *Sample et al., 2012*; *Zippin et al., 2004*). While the necessity of a proper nuclear PKA activity for neuronal function is well-documented, neither the existence of PKA R-subunits nor their physiological role within the nucleus has been well-studied.

In this study, we generated high-resolution large-scale mosaic images of several mouse brain slices using RI$\beta$- and RII$\beta$-specific antibodies. As RI$\beta$ is the least-studied isoform at the protein level, we focused our analyses on brain regions where we expected RI$\beta$ to be predominant on the basis of its mRNA expression profiles and its predicted functional importance from RI$\beta$(–/–) mice. The use of large-scale immunohistochemical brain maps allows us to gain an overview of the RI$\beta$ and RII$\beta$ protein distributions over large areas and then to zoom in to obtain higher-resolution views in order to investigate subcellular features. We found that each regulatory isoform is predominant in distinct brain regions and were able to identify unique and consistent patterns of distribution within the

hippocampus and the cerebellum. RIβ is concentrated in dendrites, and co-localizes with MAP2, whereas RIIβ is concentrated in axons. We confirmed the RIβ subcellular distribution that emerged from the mosaic images using the mini-Singlet Oxygen Generator (miniSOG), a probe that allowed us to do correlated light and electron microscopy. We found RIβ in the mitochondria, as we predicted earlier, as well as in the nucleus, establishing a new paradigm for PKA signaling in the nucleus. To demonstrate a functional distinction between the two isoforms in the nucleus, we selectively downregulated the two isoforms and used CREB phosphorylation as a reporter for a nuclear PKA substrate. Downregulation of RIβ, but not of RIIβ, decreased pCREB in hippocampal cultures. These comprehensive brain images are accessible to browse or download via the Cell Centered Database (CCDB).

## Results

### Overview of the regional distribution of RIβ and RIIβ across brain regions

The overall patterns of RIβ and RIIβ protein distribution across brain regions within a full coronal slice are shown in *Figure 1*. Images were acquired using confocal microscopy. Image stacks were then knit together to create high-resolution 2D large-scale brain images. The resulting image mosaics provide detailed views of the cellular and subcellular distribution of RIβ and RIIβ without losing the context of the tissue. To map the protein distribution of RIβ and RIIβ, we stained mouse brain sections with specific RIβ and RIIβ antibodies. Although RIIβ antibody is commonly used and its specificity has been previously validated (*Weisenhaus et al., 2010*), RIβ immunohistochemical imaging has not been possible previously due to the high cross reactivity of existing antibodies with RIα. Thus, it was essential to confirm the absolute specificity of the RIβ antibodies. To achieve high specificity, we removed any RIα cross-reacting antibodies with immobilized RIα protein as described in the Material and methods, and then extensively validated the specificity and selectivity of the RIβ and RIIβ antibodies (*Figure 1—figure supplement 1*) Western blots confirmed the specificity of the antibodies for RIα vs. RIβ purified proteins (*Figure 1—figure supplement 1*, upper left panel), while dot blots of the four purified R-subunit isoforms demonstrated the specificity and selectivity of our RIβ and RIIβ antibodies to the proteins in their native conformation (*Figure 1—figure supplement 1*, upper right panel). To confirm specificity in cells, we overexpressed RIβ-MKO2 or RIα-MKO2 in mouse 10 T1/2 cells; RIβ antibody detects overexpressed RIβ-MKO2 but not the overexpressed RIα-MKO2. Similar experiments were also carried out for the RIIβ antibody in order to validate its specificity and selectivity (*Figure 1—figure supplement 1*). The antibody did not crossreact with RIIα, its closest homolog. Staining the overexpressed cells with secondary antibodies alone resulted in no signal at similar levels or higher laser power, demonstrating no background from secondary antibodies in our method (*Figure 1—figure supplement 1 A4 and C4*). For an additional negative control, RIβ antibody was incubated with purified full-length human RIβ protein. The pre-absorbed antibody blocked the antibody's specific binding and no signal was detected when tested on the RIβ-MKO2 overexpressed cells or on a brain slice (*Figure 1—figure supplement 1 A5 and C5*, *Figure 1—figure supplement 2*).

These mosaic maps allow us to identify the predominant isoform within different brain regions across the sections, as defined by the brain atlas (*Paxinos et al., 2001*). We find that the overall patterns of RIβ and RIIβ distribution are discrete. As shown in *Figure 1A*, RIβ is expressed predominantly in the hippocampus, specifically in the CA1–3 pyramidal cell layers as well as in the dentate gyrus. RIβ is also highly expressed at the medial habenular nuclei but not at the lateral habenular nuclei of the forebrain. Strong signals are observed in the allocortex, including the piriform cortex and the retrosplenial granular cortex, and much less in the neocortex. RIβ is observed also in the dorsal endopiriform nucleus. In the hypothalamus, strong cell body labeling is seen in the subincertal nucleus. Comparing RIβ protein levels to previously reported RIβ mRNA levels (*Cadd and McKnight, 1989*), we observe that RIβ mRNA levels are higher in the pyramidal cell layer than in the dentate gyrus but the protein is expressed at similar levels in both regions.

In our images, RIIβ is predominantly expressed in the striatum (*Figure 1B*). RIIβ was previously shown to be the major PKA isoform in the striatum — including the caudoputamen, nucleus accumbens, and islands of calleja — by in situ hybridization analysis (*Cadd and McKnight, 1989*), as well

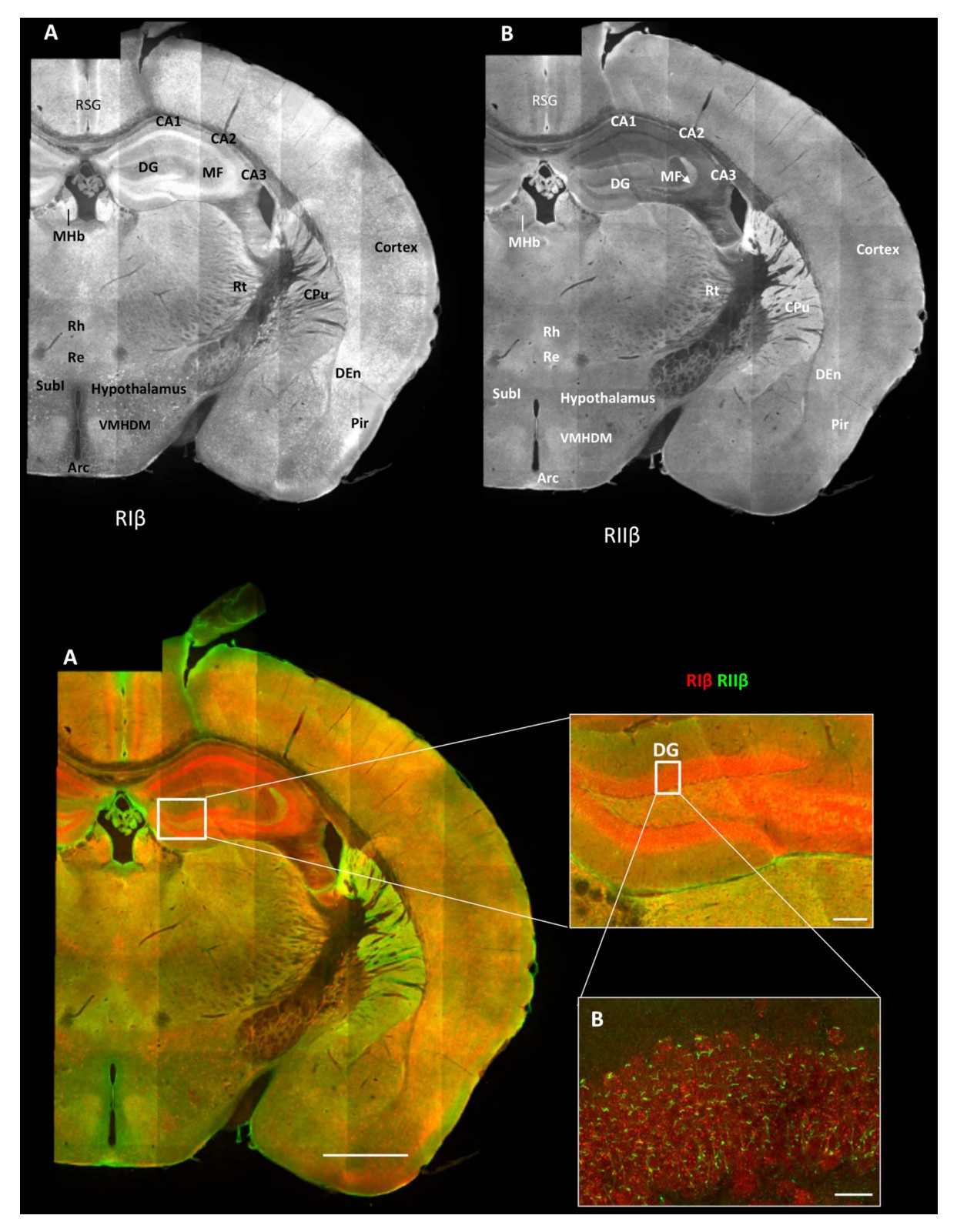

**Figure 1.** Overview of the regional distribution of RIβ and RIIβ across brain regions. Wide-field views of a high-resolution large-scale mosaic image for a full coronal tissue slice are shown for RIβ (red in colored image) and RIIβ (green in colored image). The tissue section was labeled with anti-RIβ- and anti-RIIβ-specific antibodies. The mosaic image was made up of 105 tiles. Each tile is a maximum intensity projection of a stack of 15 Z-sections that were stitched together to reconstruct this single, high-resolution 2D image. Colored image: (C) Scale bar inside the full mosaic image: 1 mm. Scale bar

*Figure 1 continued on next page*

*Figure 1 continued*

inside the small box: 100 μm. (**D**) Intermediate resolution sample. (**E** Full-resolution sample of the mosaic allows examination at higher magnification. White boxes represent the areas from which the image was captured. DG, Dentate Gyrus. Scale bar: 10 μM. Abbreviations: Arc, arculate nucleus; CPu, Caudate putamen; DEn, dorsal endopiriform nucleus; MF, mossy fibers; MHb, medial habenular nuclear; Pir, piriform cortex; Re, reuniens thalamic nu; Rh, rhomboid thalamic nucleus; RSG, retrosplenial granular cortex; RSV, retrosplenial granular; Rt, reticular thalamic nucleus; SubI, subincertal nucleus.

The following figure supplements are available for figure 1:

**Figure supplement 1.** Antibody specificity.

**Figure supplement 2.** Specificity of secondary antibodies.

as by comparing the RIIβ protein expression from various dissected brain regions (*Brandon et al., 1998*). RIIβ is also seen strongly at the choroid plexus and in blood vessels. RIIβ strongly labels the mossy fibers of the hippocampus but not the pyramidal cell layer of the hippocampus. Increased labeling within specific nuclei of the thalamus and hypothalamus was observed. On the basis of comparison with the Franklin and Paxinos mouse brain atlas, we posit that substantial levels of RIIβ are also evident at the reticular thalamic nucleus, rhomboid and reuniens thalamic nucleus. In the hypothalamus, ventromedial nuclear cells are strongly labeled as was the arculate nucleus in the mediobasal hypothalamus.

## Differential expression of RIβ and RIIβ at the hippocampus

To investigate the differential expression pattern of RIβ and RIIβ in detail within any specific region of interest, we used the brain images to zoom in from the gross sectional levels to the sub-neuronal levels of a specific region of interest. The images provide detailed views of cellular and subcellular structures without losing the context of the brain region. The problem of relating more magnified views with the lower-scale surveys is eliminated in this method. *Figure 2A* shows a detailed view of the hippocampus. RIβ is specifically localized to the pyramidal cell layers of CA1–3 region. RIβ is localized to the cell bodies of these cells as well as to their dendrites, including both basal dendrites and the apical dendrites (*Figure 2B*, upper panel). Interestingly, RIβ, but not RIIβ, is localized to the hippocampal interneurons, which are the inhibitory neurons (*Figure 2B*, lower panel). RIIβ is excluded from the pyramidal cell layer (*Figure 2C*). The protein-expression patterns seen in these maps correlate well with the mRNA detected by in situ hybridization (*Cadd and McKnight, 1989*).

## RIIβ is concentrated in axons, whereas RIβ is concentrated in dendrites and somata in various subfields of the hippocampus

We next utilized the large-scale mosaic images to explore detailed views of various subfields within the hippocampus. The advantage of the high-resolution large-scale mosaic images is that these comprehensive maps make it possible to analyze protein localization at gross structural level while at the same time providing subcellular details, thereby allowing the tracing of an axonal pathway and at the same time visualizing details such as axon terminals (*Figure 3—figure supplement 1*). We can trace an axonal pathway, in which RIIβ is enriched, that emerges from the hilus of the dentate gyrus to the mossy fibers (*Figure 3A*, *Figure 3—figure supplement 1*). RIIβ is abundant in the axons that emerge from the granule cells of the dentate gyrus and pass through the hilus (*Figure 3B*). RIIβ is also concentrated in the mossy fiber axon terminals in the CA3-stratum lucidum (*Figure 3C*). This is consistent with in situ hybridization data that show that the granule cells of the dentate gyrus express RIIβ mRNA abundantly (*Cadd and McKnight, 1989*). RIβ localization is distinct in these regions as it is concentrated in the CA3 pyramidal cell layer including the dendrites. RIIβ is abundant in the axons of the stratum lacunosum-moleculare layer, which contains perforant path fibers (*Figure 3D*). The advantage of the detailed hippocampus map is that we can zoom in into different axons within the subfields, such as the stratum lacunosum-layer or stratum radiatum, in order to identify synaptic boutons, which helps us to assign the staining specifically to axons (*Figure 3—figure supplement 1*). RIIβ is also enriched in the alveus, which is the deepest layer that contains the axons from pyramidal neurons, which pass on towards the fimbria/fornix, one of the

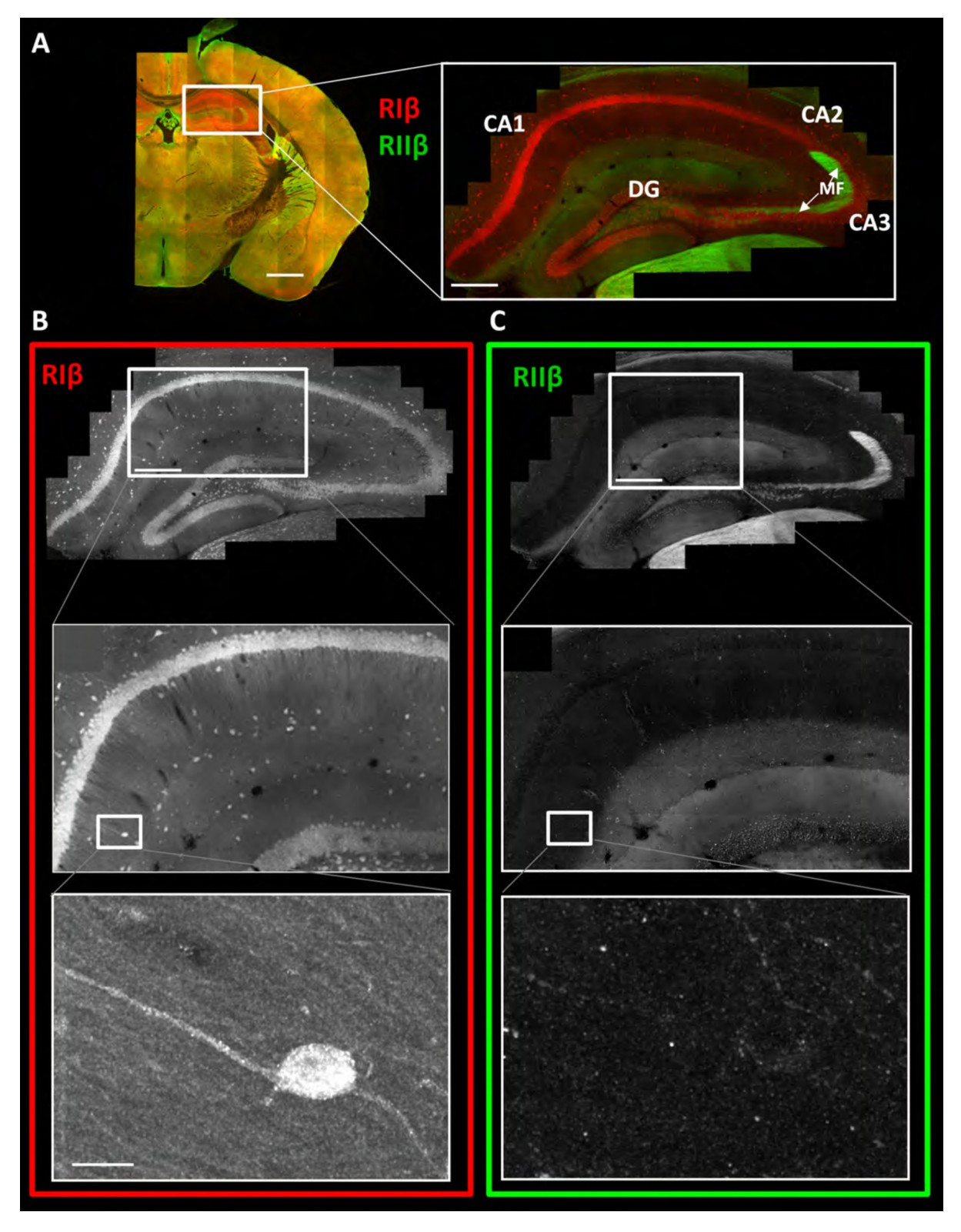

**Figure 2.** RIβ is predominantly localized to cell bodies and dendrites in different hippocampal cell types. (**A**) Coronal sections were taken from a mouse brain and labeled with anti-RIβ (red) and anti-RIIβ antibodies (green). Left: mosaic image of a full brain section made up of 105 tiles. Scale bar: 1 mm. Right: mosaic image of the hippocampus region made up of 1,413 tiles from ten Z sections obtained using a 60x objective lens. Scale bar: 200 μm. (**B–C**) Top: Mosaic image of the hippocampus at reduced resolution with anti RIβ (**B**) or anti RIIβ (**C**). The white box represents the area from which the

*Figure 2 continued on next page*

*Figure 2 continued*
middle image was captured. Middle: Higher resolution sample of the mosaic showing the pyramidal cell layer in the CA1-2 subfield. Anti- RI$\beta$ (left) or anti-RII$\beta$ (right). The white box represents the area from which the bottom image was captured. Scale bar: 200 µm. Bottom: A full resolution image of the mosaic shows a representative interneuron cell within the hippocampus region. Anti-RI$\beta$ (left) or anti-RII$\beta$ (right). Scale bars: 10 µm.

major outputs of the hippocampus (*Figure 3E*). The parallel path of axonal fibers, immunostained by RII$\beta$, in the alveus is evident (*Figure 3—figure supplement 1*). RII$\beta$ axonal staining does not co-localize with the RI$\beta$ dendritic staining. RI$\beta$ is excluded from these axons.

These data are consistent with the RII$\beta$ enrichment seen at the mossy fibers (*Weisenhaus et al., 2010*), but is in contrast to the data provided by Zhong et al (*Zhong et al., 2009*) who analyzed RII$\beta$ localization at high resolution in dendritic shafts and spines using both cultured neurons and an in vivo approach using transfected PKA subunits.

## RI$\beta$ and RII$\beta$ distinct expression at the cerebellum

We generated additional maps to examine the overall protein distribution of RI$\beta$ and RII$\beta$ isoforms in the cerebellum and to provide a full view of one folium (*Figure 4*). We focused on the cerebellum because RI$\beta$, the least-studied PKA isoform, is the most predominant PKA isoform in this region and because no immunohistochemical studies had been published previously on this isoform. Western blot analysis of the cerebellum fraction showed that RI$\beta$ was expressed in this region (*Grönholm et al., 2003*) whereas RII$\beta$ expression was not detected (*Weisenhaus et al., 2010*), probably because of its low expression compared to that detected in other regions. The in situ hybridization data showed much less hybridization of RII$\beta$ compared to RI$\beta$ at the cerebellum, which is consistent with the protein expression (*Cadd and McKnight, 1989*). A detailed comparison of the two isoforms shows distinct localization patterns. RI$\beta$ is highly enriched in the Purkinje cells, including the somata and the dendritic trees (*Figure 4B–C*). RI$\beta$ is also localized to the granular cell layer and excluded from the white matter (*Figure 4D*). RII$\beta$ is preferentially expressed at the somata of the Purkinje cells and is detected at lower levels in their dendrites, and mainly in the primary branches. The RII$\beta$ expression is consistent with previous reports as well as with the localization of AKAP150, which is the primary AKAP responsible for targeting PKA to dendritic spines and post-synaptic density (*Glantz et al., 1992*). In comparison with RII$\beta$, RI$\beta$ is abundant at the cerebellar glomerulus (*Figure 4D*).

Consistently, the full mosaic maps of the cerebellum and the hippocampus demonstrate that dendrites preferentially express the RI$\beta$ isoform at higher levels. RI$\beta$ is expressed at the dendrites of the pyramidal cell layer in the hippocampus as well as at the dendritic trees of the Purkinje cells at the cerebellum. RII$\beta$ expression was excluded from the pyramidal cell layer and was very weak at the dendrites of the Purkinje cells, suggesting a unique role for RI$\beta$ at cell dendrites.

## RI$\beta$ co-localizes with MAP2 at the cerebellum

The subcellular localization of PKA regulatory subunits is typically controlled by A kinase anchoring proteins (AKAPs) (*Wong and Scott, 2004*). Microtubule-associated protein 2 (MAP2), a dendritic marker, is also known as a dendritic AKAP that plays a substantial role in type II PKA localization in dendritic shafts (*Theurkauf and Vallee, 1982*). Since RI$\beta$ is concentrated in dendrites and abundantly expressed in the cerebellum compared to RII$\beta$, we performed co-localization experiments with MAP2 in this region. *Figure 5A* shows that in cerebellum slices RI$\beta$ co-localizes with MAP2. The RII$\beta$ isoform, which has been shown to interact with MAP2 in the striatum and hippocampus, is less concentrated in the cerebellum and co-localizes with MAP2 only at the dendritic branches of the Purkinje cell (*Figure 5B*). In hippocampal/cortical primary cultured cells, we show that RI$\beta$ co-localizes with MAP2, and consistent with previous reports, RII$\beta$ also partially co-localizes with MAP2 in these cells (*Figure 5C–D*). Although more experiments need to be done to verify direct interactions between RI$\beta$ and MAP2, our study suggests that MAP2 may contribute to RI$\beta$ anchoring to dendrites.

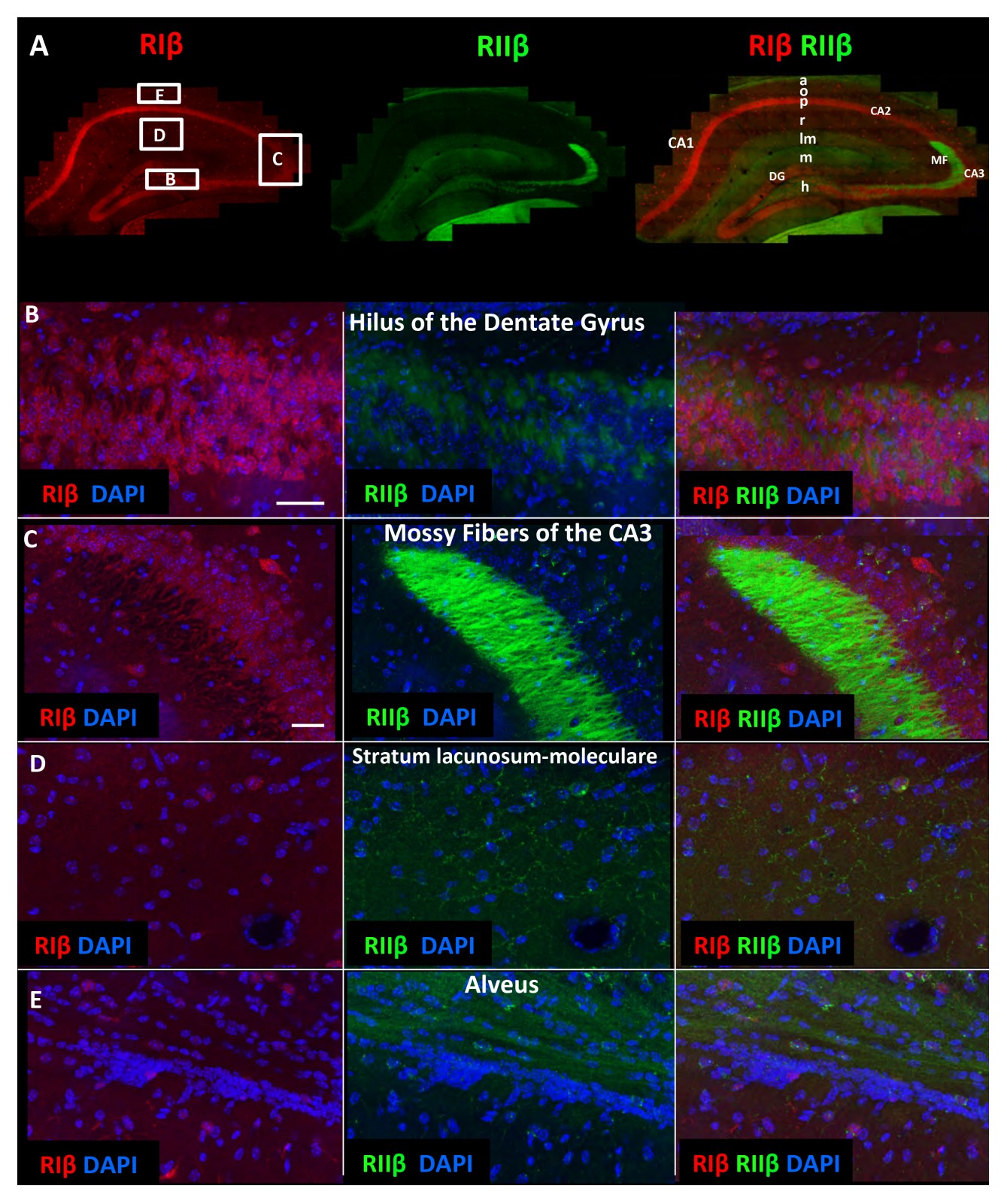

**Figure 3.** RIIβ is the predominant isoform in cell axons across various hippocampal subfields. (**A**) A full representative mosaic of a coronal section through the dorsal hippocampus at reduced resolution. An overview of RIβ (red), RIIβ (green) and RIβ/RIIβ immunostaining at various subfields. The mosaic image is made up of 1,413 tiles from ten Z sections obtained using a 60x objective lens. The white box represents the area from which the (**B–E**) subfield full-resolution images were captured. (**B**) Hilus of dentate gyrus. (**C**) Mossy fiber of the CA3. (**D**) Stratum lacunosum-moleculare. (**E**) Alveus.

*Figure 3 continued on next page*

*Figure 3 continued*

Abbreviations: CA1, stratum oriens; g, granule cell layer (stratum granulosum); h, hilus proper; lm, stratum lacunosum-moleculare; m, dentate molecular layer (stratum moleculare); p, stratum pyramidale; r, stratum radiatum. Scale bar: 25 μm.

The following figure supplement is available for figure 3:

**Figure supplement 1.** Mosaic images of the hippocampus allow tracing of RIIβ along an axonal pathway and visualization of axonal boutons in the full-resolution mosaic image.

## RIβ is a nuclear PKA isoform in neuronal cells

While the translocation of the catalytic subunit into the nucleus and the phosphorylation of nuclear factors have been well characterized, the function of nuclear regulatory subunits remains largely unexplored. In our brain maps, we typically find RIβ in the nucleus whereas RIIβ is excluded from the nucleus. Examples of the RIβ nuclear localization are shown in *Figure 6*, and include the Purkinje cells of the cerebellum (*Figure 6A*), interneurons in the hippocampus (*Figure 6B*), and neuronal cells in the hypothalamus (*Figure 6C*). We confirmed that the secondary antibodies do not provide non-specific background on the brain slice (*Figure 6—figure supplement 1*). In primary cultured cells of hippocampal/cortical neurons, RIβ is also found at the cell nucleus, whereas RIIβ is excluded from the nucleus (*Figure 6D*). Stimulation of these cells with forskolin to activate adenylate cyclases did not result in RIIβ translocation to the nucleus (data not shown). The molecular processes that direct translocation of the R-subunits to the nucleus are as yet undefined.

## RIβ subcellular localization identified by electron microscopy

To further assess and to define more precisely the nuclear localization of RIβ observed by the fluorescent images, we used electron microscopy and the recently introduced miniSOG labeling (*Shu et al., 2011*). MiniSOG is a genetically modified flavoprotein, which produces singlet oxygen upon excitation. DAB will then be oxidized to form an insoluble osmium-philic polymer, which gives contrast to the ultrastructure near the tagged fusion protein. RIβ cDNA was fused to miniSOG and electrophorated into primary hippocampal cells. As shown in *Figure 7D*, we found that RIβ is localized to the mitochondria, as we previously showed using cell fractionation experiments (*Ilouz et al., 2012*). The resolution allows us to localize RIβ more specifically to the cristae and the inner membrane of the mitochondria. A control untranfected photo-oxidized mitochondrion is shown in *Figure 7C*. 126 transfected darkened mitochondria and 71 non-transfected light mitochondria were counted. Representative images for a non-transfected (*Figure 7C*) and a transfected (*Figure 7D*) mitochondrion are shown. Moreover, we can detect RIβ localization in the cell nucleus, confirming the fluorescent images of brain sections (*Figure 7B*). A control untransfected photo-oxidized cell nucleus is shown in *Figure 7A*. Consistent with this, RIβ protein levels are much lower in the human brain cytosolic fraction than in the particulate fraction. It is only the RIβ regulatory subunit, among all four regulatory subunits, that shows high protein levels in the particulate fractions of postmortem brains (*Chang et al., 2003*).

## RIβ, but not RIIβ, contributes PKA-dependent CREB phosphorylation

One of the nuclear targets of PKA is the transcription factor cyclic AMP response element-binding protein (CREB). The necessity for PKA activation and a subsequent phosphorylation of CREB on Ser133 is well documented for many neuronal signaling pathways, including synaptic plasticity (*Kandel, 2012*), but the specific physiological functions of PKA regulatory subunits in these signaling complexes are only beginning to be understood. As we found RIβ in the nucleus, we sought to investigate whether downregulation of RIβ would result in impairment of CREB phosphorylation, a nuclear PKA substrate, in cultured cells. To accomplish this, we designed lentiviral shRNA vectors containing GFP to knockdown RIβ or RIIβ subunits. Thus, cells expressing shRNA plasmids can be identified by the presence of GFP. We first confirmed the knockdown efficiency by co-transfecting 293T cells with cDNA encoding either RIβ or RIIβ genes, together with a scrambled shRNA (shRNA IRR) as a negative control or shRNAs targeting either RIβ or RIIβ. Western blot analysis confirmed the knockdown efficiency for RIβ at the protein level (*Figure 8—figure supplement 1A*). Western

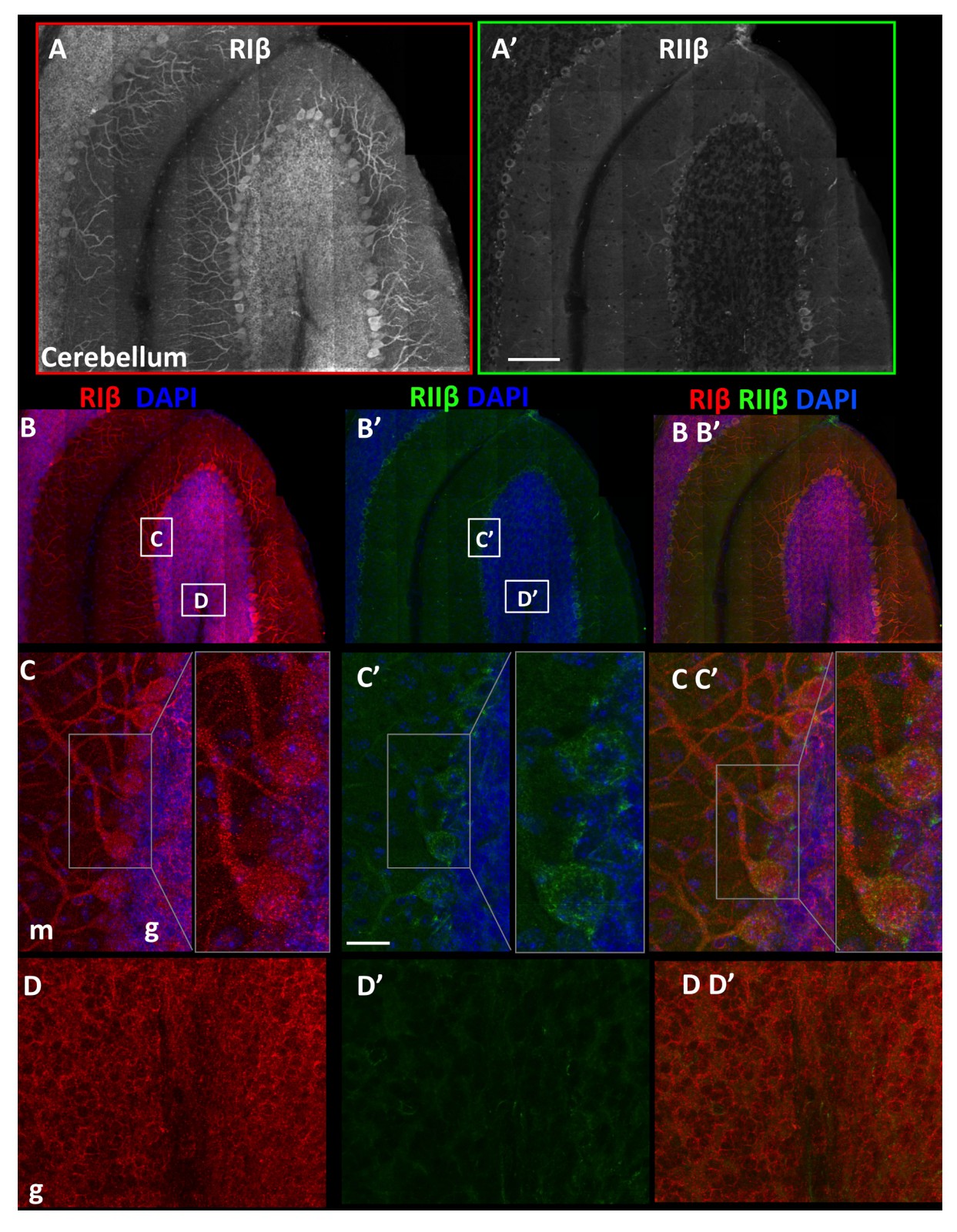

**Figure 4.** RIβ and RIIβ distinct expression at the cerebellum. **(A)** An overview of a full representative sagital section of one folium of the cerebellum immunostained with anti-RIβ (**A**) or anti- RIIβ (**A'**) at reduced resolution. The mosaic image is made up of 120 tiles, from 21 Z sections obtained using a 40x objective lens. Scale bar: 100 μm (**B**). RIβ (red), RIIβ (green) and Dapi (blue). White boxes represent the areas from which (**C–D**) images were captured. **(C)** Cerebellar Purkinje cells at the molecular layer (m) and the granular layer (g) are indicated. Right side: full-resolution views of Purkinje

*Figure 4 continued on next page*

*Figure 4 continued*

cells. RI$\beta$ is localized to the somata and the dendrites of these cells. RII$\beta$ is less abundant at their dendrites. Scale bar: 25 µm. (**D**) Full-resolution views taken from the granular cell layer (g) and the white matter.

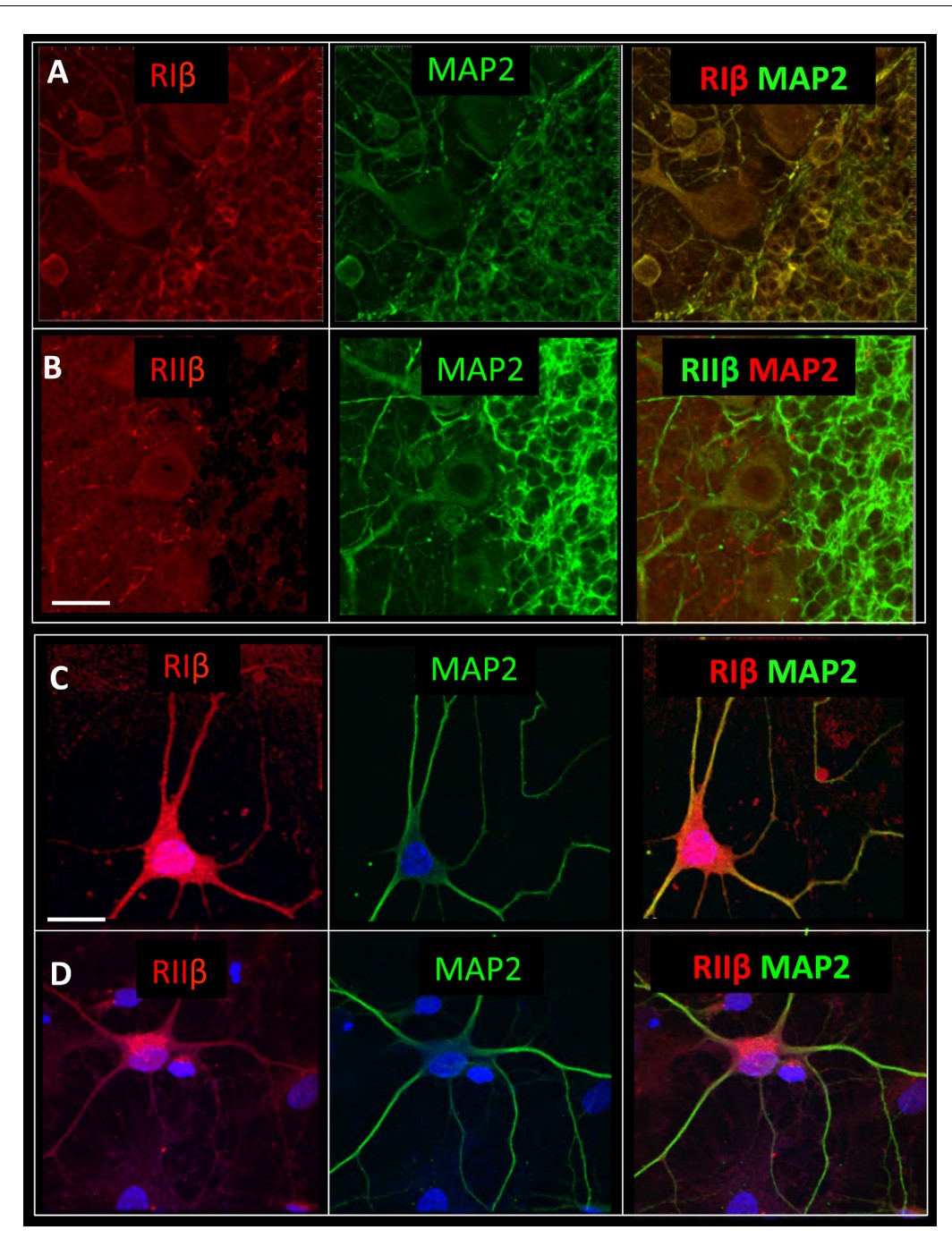

**Figure 5.** RI$\beta$ co-localizes with MAP2. Sagital sections of the cerebellum were co-stained with microtubule-binding protein (MAP2) antibody, a dendritic marker, and (**A**) a RI$\beta$ antibody or (**B**) a RII$\beta$ antibody. Primary cortical hippocampal cultured cells were co-stained with (**C**) MAP2 antibody and RI$\beta$ antibody or (**D**) RII$\beta$ antibody. Dapi staining for the nucleus is shown in blue. Scale bars: 20 µm.

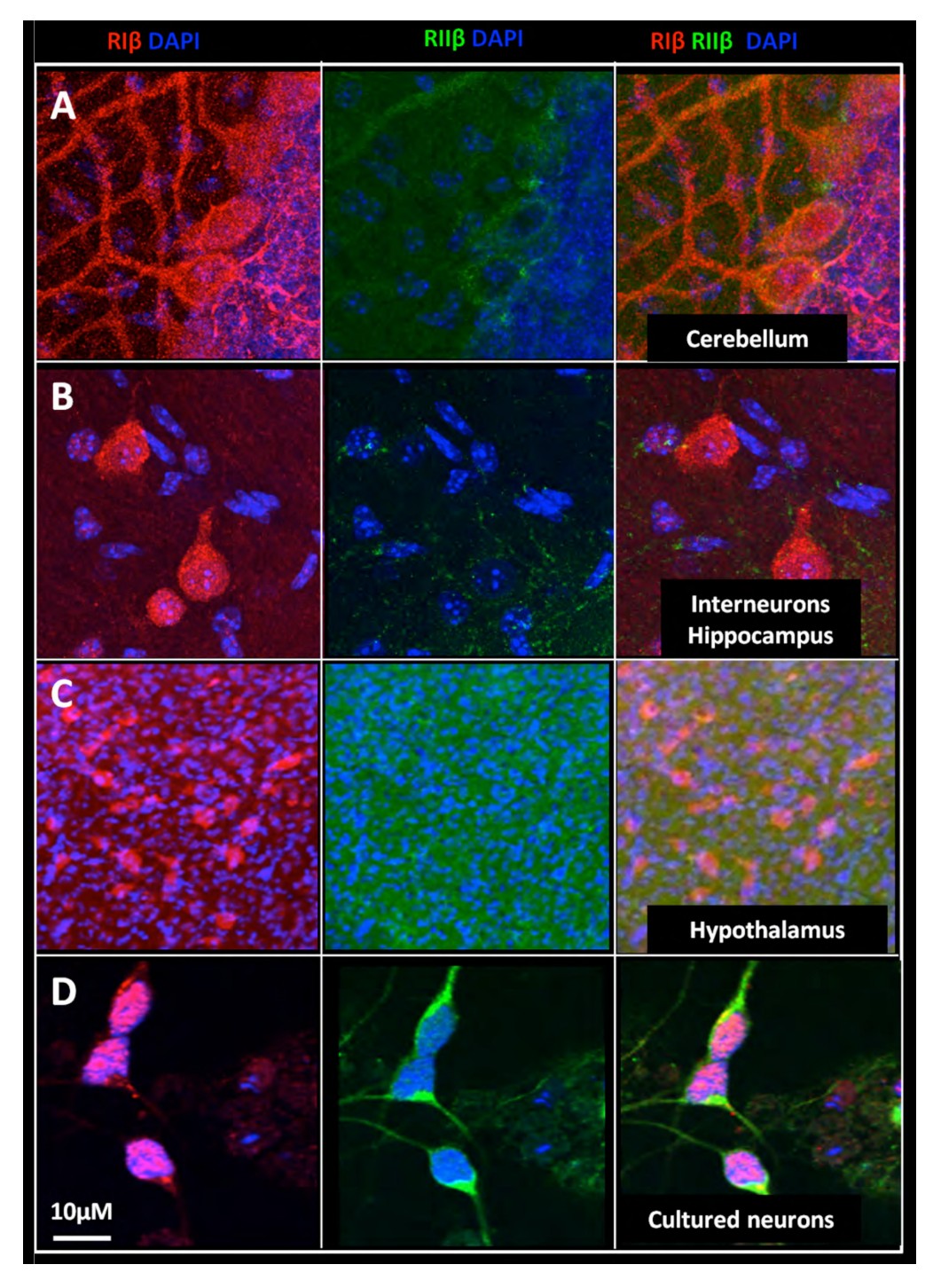

**Figure 6.** Nuclear localization of RIβ subunit across various brain regions. Full-resolution views of different brain regions were taken from the mosaic maps or primary cell staining to show the nuclear co-localization. RIβ (red), RIIβ (green) and dapi (blue). One slice of the Z stack is shown (**A**. Purkinje cells of the cerebellum. (**B**) Interneurons in the hippocampus. (**C**) Cells from the thalamus. (**D**) Cells from the hypothalamus. (**E**) Primary cortical/hippocampal cells. Scale bar: 10 μm.

The following figure supplement is available for figure 6:

*Figure 6 continued on next page*

*Figure 6 continued*

**Figure supplement 1.** Tissue staining with secondary antibodies alone shown as negative controls for brain regions indicated in *Figure 6*.

blot analysis indicated that an efficient RIIβ knockdown had been achieved using shRNA RIIβ clone #2 (*Figure 8—figure supplement 1B*). This clone was used in the subsequent experiments to knock-down RIIβ. To further validate the knockdown efficiency of these shRNA lentiviruses in primary cultured neurons,we infected the cells with shRNAs targeting either RIβ or RIIβ and used scrambled shRNA (shRNA SCR) as a negative control. Cells were immunostained with either RIβ- or RIIβ-specific antibodies. Infected cells expressing the indicated shRNAs were identified by the presence of GFP. Cells expressing shRNA against RIβ or RIIβ display a significant reduction in the cell fluorescence intensity produced by their specific antibodies as compared to cells infected with a negative control. The results indicate a significant inhibition of RIβ or RIIβ expression in cells infected with shRNA against a corresponding gene. Representative cells are shown in *Figure 8—figure supplement 1C*.

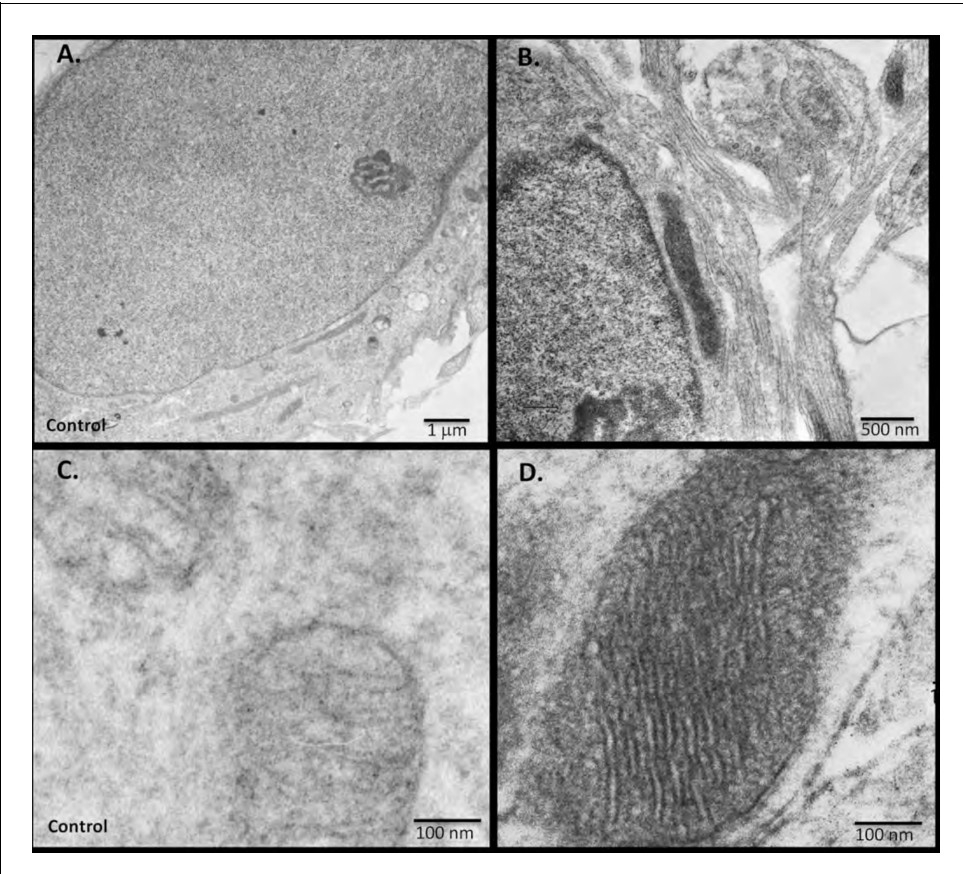

**Figure 7.** miniSOG-tagged RIβ localizes RIβ to the mitochondria and to the nucleus using electron microscopy. Primary hippocampal or cortical cells were electrophorated with miniSOG-tagged RIβ. Mersalyl acid was used in the blocking step to reduce background nonspecific labeling of mitochondria. The differential contrast generated between (**A**) a non-transfected nucleus and (**B**) a transfected nucleus following photooxidation is evident. RIβ is localized to the nucleus and nuclear envelope. The differential contrast generated between (**C**) a non-transfected photo-oxidized mitochondrion and (**D**) a transfected photo-oxidized mitochondrion is evident. The well-preserved mitochondrion allows detection of RIβ at the mitochondrial cristae and the inner membrane. 126 transfected darkened mitochondria and 71 non-transfected light mitochondria were counted. Representative images for (**C**) non-transfected and (**D**) transfected mitochondria are shown.

The shRNAs against RI$\beta$ or RII$\beta$ demonstrate 70% or 80% knockdown efficiency, respectively, in neuronal cells (p-value <0.0001) (*Figure 8—figure supplement 1D*).

Following knockdown validation, we sought to investigate whether downregulation of RI$\beta$ would result in the impairment of CREB phosphorylation. We infected dissociated rat hippocampal cultures with GFP-shRNA lentiviruses against RI$\beta$, RII$\beta$ or an irrelevant control. After recovery from infection, cells were stimulated with forskolin, an adenylyl cyclase activator, to increase intracellular levels of cAMP and to activate PKA. The physiological consequence of RI$\beta$ and RII$\beta$ downregulation for nuclear cAMP signaling was evaluated by analysis of CREB phosphorylation at Ser 133. Owing to the robust survival capacity of non-neuronal cells, trace gila present in initial preparations were an unavoidable component of these cultures. Non-infected cells remain as well. To specifically assay CREB phosphorylation in infected cells, the signal was quantified only in cells that were positively stained for $\beta$III-tubulin (neuronal marker, red) and that expressed GFP (the marker for lentivirus infection). CREB phosphorylation was quantified in the specified cells as a measure of pCREB signal intensity per the area of the nucleus as delineated by DAPI. We find that in forskolin-stimulated neuronal cells, a significant reduction in the intensity of phospho-Ser133 CREB was detected in neurons infected with shRNA against RI$\beta$ compared to the control (p<0.0001) (*Figure 8A*, *B*). B contrast, no significant decline in the intensity of phospho-Ser133 CREB was detected in cells infected with shRNA against RII$\beta$ (*Figure 8A*, *B*). The decline in phospho-CREB in neurons infected with shRNA against RI$\beta$ was also significant compared to neurons infected with shRNA against RII$\beta$ (p=0.0003).

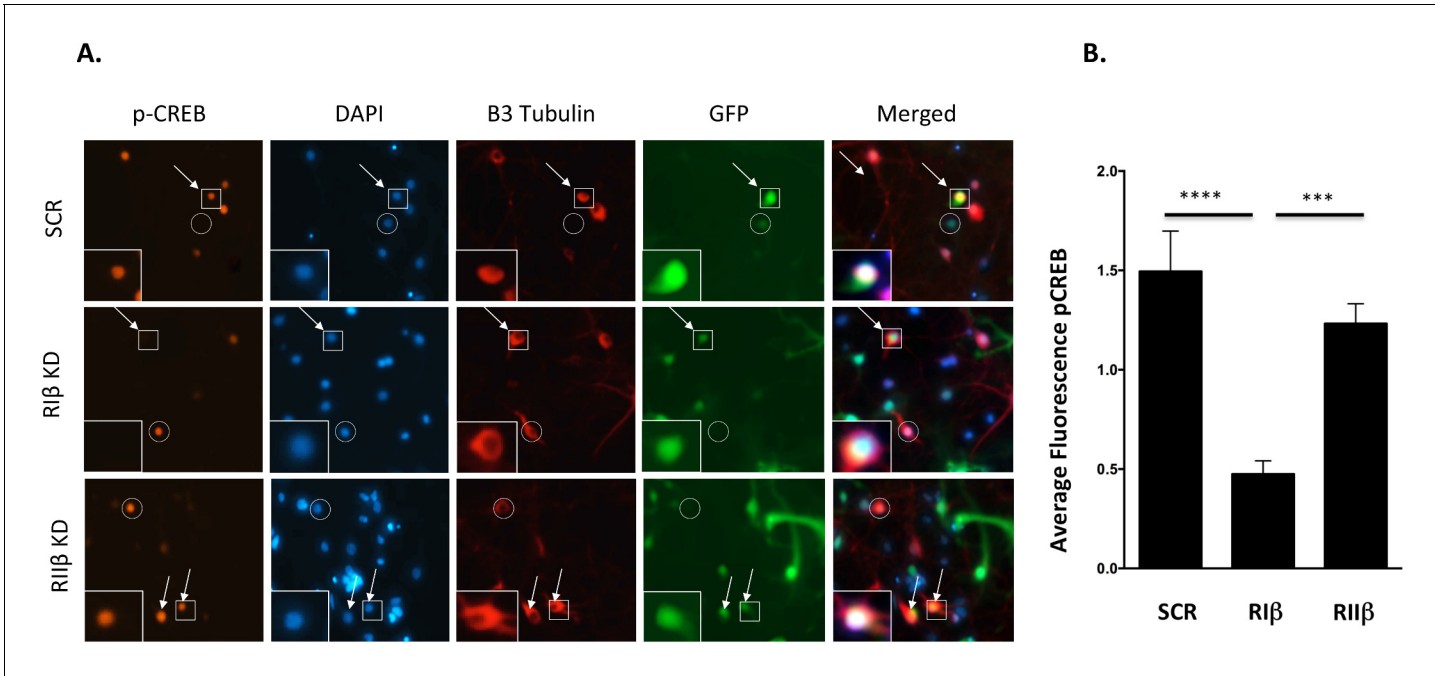

**Figure 8.** Downregulation of RI$\beta$ in neurons reduces CREB phosphorylation. (**A**) Primary hippocampal cells were treated with shRNAs against RI$\beta$, RII$\beta$ or an irrelevant control. CREB phosphorylation was induced by forskolin incubation. Intensity of CREB phosphorylation at Ser133 was quantified in the nucleus (delineated by DAPI) of neuronal cells (delineated by $\beta$III tubulin) that were infected by the GFP-shRNA lentiviruses (representative cells are indicated by arrows). Either non-neuronal GFP-expressing cells or neuronal cells that did not express GFP were excluded from the analysis (examples are indicted by a circle). Representative infected neurons depicting p-CREB, DAPI, $\beta$III-tubulin and GFP (in squares) are shown close up. (**B**) Relative fluorescence of pCREB was quantified for each condition using imageJ. Statistical analyses were performed using Tukey's multiple comparisons test under one-way-ANOVA (GraphPad Prism6 software). Adjusted p-values are reported. In the RI$\beta$ knockdown, there is a significant reduction in pCREB fluorescence intensity when compared to control (****p<0.0001) or to RII$\beta$ (***p=0.0003). There is no statistical significance between control and RII$\beta$ (p=0.3999). Results were assessed from three independent experiments. ~50 cells were quantified for each condition.

The following figure supplement is available for figure 8:

**Figure supplement 1.** Validation of shRNA knockdown efficiency.

## Discussion

Signaling events mediated by PKA are critical for many neuronal functions. While the necessity of proper PKA catalytic activity and the phosphorylation of its downstream substrates are well established in a variety of neuronal functions, the rational and significance for having multiple PKA regulatory isoforms has not been appreciated. In this study, we used highly specific antibodies against RI*β* and RII*β* to create high-resolution large-scale mosaic images of the brain, and these images revealed distinct patterns of localization. Although RI*β* is expressed at low levels in most cells, it is enriched in the brain, and our images show that RI*β*, but not RII*β*, is highly expressed in the hippocampus, consistent with tissue fractionations and with many of its predicted roles in neuronal function. In spite of its importance for neuronal function, RI*β* has never been comprehensively imaged before due to the lack of specific antibodies. Furthermore, there has never been a systematic comparison of these two isoforms that are abundantly expressed in the brain. The mouse brain mosaic images presented here thus provide detailed global comparative views of whole brain sections and also help to explain the functional non-redundancy of RI*β* and RII*β*. In addition, the maps allow the interrogatation of subcellular features without losing the context of the brain regions. By zooming in to different brain regions, such as the hippocampus and cerebellum, we consistently find that RI*β* is enriched in dendrites and co-localizes with MAP2, whereas RII*β* is more concentrated in axons. In our mosaic maps, RI*β* is also localized to the nucleus within different brain regions. To verify the subcellular localization at higher resolution, we used miniSOG labeling, a technique that allows us to correlate light microscopy with electron microscopy. Consistent with our earlier report, we found RI*β* at the mitochondria (*Ilouz et al., 2012*), and with the miniSOG labeling, we could further localize RI*β* to the inner membrane and cristae of the mitochondria. In addition, RI*β* was prominently seen in the nucleus and around the nuclear envelope. To demonstrate a functional distinction between the two isoforms at the nucleus, we used primary neuronal cells and specific lenti-viruses to downregulate RI*β* or RII*β* selectively and then tracked CREB phosphorylation as a reporter for a PKA nuclear substrate. Downregulation of the RI*β*, but not of RII*β*, decreased CREB phosphorylation, which is consistent with the observed nuclear localization of RI*β*.

Our mosaic maps provide an overview of the distribution of endogenous PKA regulatory subunits in a whole brain slice, and at the same time allow us to identify subcellular features. The distinct subcellular localization of PKA RI*β* and RII*β* subunits within discrete brain regions emphasizes the importance of mapping the spatial localization of proteins in the context of the tissue, whereas cultured cells sometimes do not provide an accurate representation of spatial localization. We find that each regulatory subunit is predominant within different brain regions and has a distinct and consistent patterns of cellular and subcellular localization across multiple regions. This isoform-specific localization provides an additional layer of control for achieving specificity in PKA signaling, and identifies localized pools of PKA signaling complexes in the brain. The specific pattern of distribution of RI*β* in the hippocampus is consistent with its known unique roles in synaptic plasticity (*Brandon et al., 1995*). RI*β* but not RII*β* is predominant at the pyramidal neurons in the CA1–CA3 regions of the hippocampus, as well as in the dentate gyrus and hilus neuron (*Figure 1*). This localization is consistent with hippocampal slices from RI*β* null mice, which showed a severe deficit in LTD and depotentiation at the Schaffer collateral-CA1 synapse. This defect was also evident at the lateral perforant path-dentate granule cell synapse in RI*β* knockout mice (*Brandon et al., 1995*). By contrast, and also consistent with previous reports, our maps show that RII*β* is not expressed at the pyramidal neurons but is concentrated in the striatum. In correlation with the RII*β* expression pattern, targeted disruption of the RII*β* gene in mice led to a dramatic downregulation of total PKA activity in the striatum and a partial compensatory increase in RI subunits (*Brandon et al., 1998*).

Nuclear signaling events mediated by PKA are critical for numerous neuronal functions, including the activation of gene expression that is required for long-term memory (*Kandel, 2012*). A surprising and intriguing finding in this study is that RI*β* is the PKA regulatory subunit that is localized to the nucleus of many cells in different brain regions, and miniSOG labeling of RI*β* confirmed the nuclear localization at a nanoscale resolution by electron microscopy. Induction of cAMP and activation of PKA leads to phosphorylation of specific transcription factors and induction of gene expression and long-term changes in neurons (*Greengard, 2001*). Nuclear localization of RI and RII has been reported in hepatocytes by immunogold labeling (*Kuettel et al., 1985*). There is evidence to suggest that nuclear RII*β* binds CREB in T cells, implicating nuclear RII*β* as a potential negative

regulator of T cell activation (*Elliott et al., 2003*). Whether PKA is resident in the nucleus or translocated into the nucleus in response to cytoplasmic signals is still an open question. Clearly one now also needs to explore all of the PKA isoforms as well as specific cell types carefully.

PKA holoenzymes are recruited into well-defined macromolecular signaling complexes through a specific interaction between the dimerization and docking (D/D) domain, a conserved hydrophobic domain within PKA R-subunits, and A Kinase Anchoring Proteins (AKAPs). These scaffold proteins determine the spatio-temporal activity of PKA. Over 50 members and splice variants of the AKAP family have been identified (*Scott et al., 2013*). While most AKAPs were initially identified by their ability to bind RII subunits, several AKAPs, such as D-AKAP1 (*Huang et al., 1997b*) and D-AKAP2 (*Huang et al., 1997a*), have been shown to bind RI as well, thereby making them dual-specific. Recently, other AKAPs such as smAKAP or GPR161 were shown to be highly specific and selective for RI subunits (*Bachmann et al., 2016*; *Burgers et al., 2012*). The first RIβ-specific AKAP, which is the only neuronal binding partner for RIβ identified to date, is the neurofibromatosis 2 tumor suppressor protein merlin (*Grönholm et al., 2003*). In this study, we provide a comprehensive comparison between the protein localization of RIβ and RIIβ in several brain regions, and show that RIβ colocalizes with MAP2 at the cerebellum. MAP2 was the first AKAP identified to bind RII subunits and tethers PKA with microtubules (*Theurkauf and Vallee, 1982*). It is only in recent years that we appreciate that some AKAPs can be dual-specific. On the basis of sequence alignment as well as structural alignment, it can be deduced that MAP2 is an AKAP that satisfies the requirement of binding both RI and RII subunits (*Sarma et al., 2010*). We thus suggest that preferential binding and interactions of either RI or RII with a potential dual-specific AKAP is likely to be cell specific and may be determined by the predominant R subunit that is expressed in a certain location.

Little is known about AKAPs that assemble and integrate RIβ holoenzyme signaling at cellular and subcellular regions; however, recent discoveries suggest that RIβ compartmentalization can indeed be linked to neurodegenerative disease. A point mutation in the dimerization and docking (D/D) domain of human RIβ, for example, has been associated with a new type of a familial neurodegenerative disease presenting with dementia and Parkinsonism (*Wong et al., 2014*). These patients also show an increase of RIβ in neuronal inclusions. This mutation can thus be classified as a targeting defect that can be validated in mouse models and in patient samples where we anticipate that RIβ localization will be altered. Although the pathophysiological mechanism by which this mutation leads to neurodegenerative disease remains to be investigated, this point mutation further emphasizes the importance of precisely controlled PKA isoform localization. Our mosaic maps, which can be downloaded and queried in great detail, represent the endogenous expression and localization of PKA isoforms. These images lay the foundation for studying biological mechanisms of neurodegenerative diseases, at subcellular and molecular levels, in which PKA regulation may be dysfunctional both in mouse models and patient samples.

## Materials and methods

### mKO2 plasmids

The mKO2-tagged PKA regulatory (R) subunits were generated by fusing respective R subunits with mKO2 to the C terminus with PCR and were inserted between EcoRI and NotI sites in pcDNA3 (Invitrogen, Carlsbad, CA). There is a SalI site as linker in RIα, RIβ, and RIIβ constructs and a BamHI site as linker in the RIIα construct.

### Antibody information

Primary antibodies: Rabbit monoclonal anti-PKA R2B Abcam catalog # AB75993 (RRID: AB_1524201) dilution 1:100. The same pattern of staining was obtained when using Anti- PKA RIIβ mouse monoclonal BD Biosciences Cat #610625 (RRID: AB_397957) at 1:200 dilution for immunofluorescence. Anti-PKA RIα Abcam catalog# ab60064 (RRID: AB_2168081) dilution 1:1000 for WB. Anti- PKA RIIα Abcam catalog number # ab38949 (RRID: AB_725890) dilution 1:1000 for WB. Mouse anti-MAP2 Sigma Aldrich M-4403 antibody at 1:200 dilution. Secondary antibodies: Cy3-Donkey anti-sheep (Jackson ImmunoResearch Laboratories) (RRID: AB_2315778), Alexa Fluor 488 Donkey anti-rabbit (Jackson ImmunoResearch Laboratories) (RRID: AB_2313584), Alexa 568 Fluor Donkey anti-mouse (Jackson ImmunoResearch Laboratories). All secondary antibodies were used at 1:250

dilution. pCREB cell signaling (RRID: AB_1658172). BIII-tubulin Covance Research Products Inc Cat# MMS-435P (RRID: AB_2313773).

## RIβ antibody purification

Sheep anti-RIβ polyclonal, R and D systems, Catalog # AF4177 (RRID: AB_2284184) was further purified as described below to eliminate cross-reactivity with RIα and used at 1:400 dilution. cAMP-sepharose resin was equilibrated with a washing buffer (20 mM MES pH 6.5, 100 mM NaCl, 2 mM EDTA, 2 mM EGTA, 5 mM DTT). 2 μg purified RIα protein was added to the resin and incubated overnight at 4°C. The next day, access RIα protein was washed from the resin using the washing buffer. Resuspended lyophilized RIβ antibody was added to the resin and incubated overnight at 4°C. The following day, resin was centrifuged at 3000 rpm for 5 min at 4°C. Supernatant, together with three resin washes with TBST, were collected and concentrated using an Amicon centrifugal concentrator. Final antibody concentration was 0.5 mg/ml.

## Dot blot experiment

Purified proteins were spotted on a nitrocellulose membrane. The membrane was incubated at room temperature for 30 min to ensure that the blots are dry. Membrane was blocked with 5% dry milk in TTBS (50 mM Tris, 0.5 M NaCl, 0.05% Tween-20, pH7.4) for 1 hr at room temperature. Membrane was incubated with primary antibodies for 1 hr at room temperature. Following 3X washes with TTBS membrane was incubated with secondary antibodies conjugated with HRP for 30 min. Following 3X washes with TTBS membrane was incubated with ECL reagent.

## Cultured cells and immunostaining

10 T1/2 and 293 T cell lines, received from research labs, tested negative for mycoplasma contamination based on DAPI staining. The 10 T1/2 cell line was not found to be on the list of commonly misidentified cell lines (International Cell Line Authentication Committee). HEK cells were found to be on the list of commonly misidentified cell lines. The authors performed no further authentication of the cell lines.

To validate antibody specificity, 10 T1/2 cells maintained in DMEM supplemented with 10% fetal bovine serum (FBS) and 2 mM GlutaMAX were grown to 80% confluency on glass coverslips in a 37°C incubator with 10% $CO_2$. Transient transfections were carried out using 2 mL of Lipofectamine reagent (Invitrogen) and 0.5 mg of DNA encoding RIβ-MKO2, RIα-MKO2, RIIβ-MKO2 and RIIα-MKO2 plasmids. Approximately 20 hr after transfection, cells were fixed with 4% paraformaldehyde. For immunostaining, cells were permeabilized with 0.3% Triton X-100 and blocked with 1% normal donkey serum, 0.5% BSA, and 50 mM glycine. Cells were stained with primary antibodies overnight at 4°C and secondary antibodies for 1 hr at room temperature. Coverslips were mounted on glass slides using ProLong Gold antifade reagent with Dapi (Invitrogen). Images were acquired using a FluoView 1000 confocal laser scanning microscope with a 60x objective lens with an NA of 1.45 (Olympus).

## Primary cultured cells

Hippocampal and cortical neurons were dissected from embryonic day 18 (E18) Sprague Dawley rats, and dissociated with papain. Cells were transfected by Amaxa electroporation (Lonza), and cultured in Neurobasal medium with B27 supplemen, 2 mM GlutaMAX, 50 U/ml penicillin and 50 μg/ml streptomycin as previously described (*Lin et al., 2008*). (Approved Animal Protocol: S03182R.)

## Subjects for mouse brain mosaic images

All experiments involving vertebrate animals conform to the National Institute of *Health Guide for the Care and Use of Laboratory Animals* (NIH publication 865–23, Bethesda, MD, USA) and were approved by the Institutional Animal Care and Use Committee (IACUC) of the University of California San Diego (Approved Animal Protocol: S03172m). In these experiments, we used male C57BL/6 mice that were approximately two months old.

## Tissue preparation and processing

Mice were fully anesthetized and PBS was perfused transcardially for 3 min followed by 4% paraformaldehyde for 10 min. The brain was removed and post-fixed in 4% paraformaldehyde overnight at 4°C. This overnight fixation step ensures better quality of vibratomed sections and reduces the number of tears in the final sections.

Sagittal cerebellum sections or coronal sections were cut on a Leica vibratome at a thickness of 75–100 microns. If not processed during the same day, tissues were stored at −20°C in cryoprotectant solution (30% glycerol, 30% ethylene glycol in PBS) until processed.

Next, free-floating sections were washed three times with 1×PBS for 5 min. Sections were then blocked with 3% normal donkey serum, 1% bovine serum albumin, 1% fish gelatin, 0.1% Triton X100, and 50 mM glycine in PBS for 1 hr at room temperature. Primary antibodies were applied overnight at 4°C. The following day, sections were washed three times with 1×PBS for 5 min and then stained with the secondary antibody for 2 hr at room temperature. Sections were washed three times with 1×PBS for 5 min before mounting on glass slides using ProLong Gold antifade reagent with DAPI (Invitrogen).

## Specimen preparation and imaging

Because wide-field brain mosaics were acquired at close to the resolution limit of light microscopy, good structural preservation and tissue quality were essential. To avoid any structural degradation associated with freezing, fixed tissue was processed unfrozen and sectioned on a vibrating microtome. Only sections devoid of tears, fold, and other defects were chosen for analysis. Care was taken during mounting and fluorescent labeling to minimize compression and to optimize staining and imaging conditions.

## Wide-scale data acquisition

A FluoView 1000 (Olympus Center Valley, PA, USA) equipped with 20x, 40x NA 1.3 oil or 60x NA 1.45 oil immersion objective and a high-precision motorized stage was used to collect the large-scale 3D mosaics of each tissue section. The boundaries (in x, y, and z) of the tissue section were defined using the Multi-Area Time Lapse function of ASW 1.7 c microscope operating-software provided by Olympus (Olympus, Center Valley, PA, USA). The software automatically generates a list of 3D stage positions covering the volume of interest, which are computed using the dimensions of a single image in microns, degree of overlap between adjacent images and z-step size. Individual image tiles were 1024 × 1024 with a pixel dimension of 0.62 μm; overlap between two adjacent images (x–y) was 10% and the z-step was ~0.5 mm/section; there is no overlap in z. The specimen was excited sequentially with a laser at two different wavelengths: 488 nm and 561 nm. The final data are stored in a RGB image volume, where the color channels map the specimen susceptibility at wavelengths 561 nm and 488 nm. Unprocessed data were stored as a single image stack containing information about the relative position of each image tile.

## Image processing

The tiles were stitched together post-data acquisition to generate the final reconstructed 2D volume using National Center for Microscopy and Imaging Research (NCMIR)-developed ImageJ Mosaic Plug-ins (RRID:SCR_001935), which was used to flat field, normalize, align, and combine images into a mosaic (*Berlanga et al., 2011*; *Chow et al., 2006*). Software is available for download (*Chow et al., 2006*). The resulting reconstructed mosaic image was downloaded and opened in Adobe Photoshop CS2 (Adobe Systems Inc., San Jose, CA, USA) (RRID:SCR_014199), which was used to to rotate, crop, and adjust color balance of the image.

## Image deposition into the cell centered database

Mosaic images were acquired using an automated imaging workflow system developed at the National Center for Microscopy and Imaging Research (NCMIR) (RRID:SCR_002655) to upload imaging data with its associated metadata directly from the microscopes in our facility and to register these datasets as microscopy products within the Cell Centered Database (CCDB). This complex system for data collection, storage, and manipulation funnels large numbers of valuable datasets directly into the CCDB (RRID:SCR_002168) (*Martone et al., 2008*). Three datasets resulting from

this publication will be released to the public through the CCDB. Researchers will be able to browse or download our large-scale mosaics through the CCDB at http://www.cellimagelibrary.org/images?k=project_20403&simple_search=Search.

Because these datasets are quite large and rich in information, each of these reconstructed mosaic images will be viewable at full resolution and annotatable without downloading using the Web Image Browser (WIB) (RRID:SCR_007015), a visualization program based on GIS technology similar to that used for Google maps, but specialized for microscopy imaging data. Through the WIB, users can turn on and off channels and adjust the contrast of large microscopic imaging datasets.

## Electron microscopy

Primary hippocampal cortical cells were transfected with miniSOG-tagged RI$\beta$ using Amaxa electroporation (Lonza). Samples were treated as described in detail previously (*Perkins, 2014*).

## Validation of shRNA lenti-viruses knockdown efficiency

293T cells were co-transfected with either PRKARI$\beta$ or PRKARII$\beta$ rat cDNA (Origene) and with scrambled shRNA (Sigma) or shRNAs targeting PRKARI$\beta$ (5'- CGGCAGAAGTCAAACTCACAGTGTGA TTC-3', purchased from Origene) or PRKARII$\beta$ (# 1–5'-CGTCATCGACAGAGGAACATT-3'; # 2–5'-CGATGCAGAGTCCAGGATAAT-3' and # 3–5'-CCTTCAGGAGAATAATAGTAA-3'). shRNA hairpins were cloned into an NheI site of the p156RRLsin lenti-vector expressing GFP (*Dull et al., 1998*). 293T cells growing on 60 mm Petri dishes were transfected with 0.5 µg plasmids encoding either RI$\beta$ or RII$\beta$ cDNA, together with 5 µg shRNA against either RI$\beta$ or RII$\beta$, or 5 µg negative control. Transfections were performed using Lipofectamin 2000 (Invitrogen) according to manufactures' instructions. Protein lysates were prepared 48 hr post-transfection using RIPA buffer supplemented with protease inhibitors (Roche). Western blot analyses carried out using RI$\beta$ or RII$\beta$ antibodies as well as actin antibody for loading control were used. Lentiviral particles were then prepared as previously described (*Ikawa et al., 2003*).

## Downregulation of RIβ or RIIβ in neurons and detection of CREB phosphorylation

Primary hippocampal neurons were isolated as described above or obtained from Life Technologies (A10840-01). Cells were plated in Neurobasal medium supplemented with 10% FBS, 1% glutamine, 2% G21, and Pen/strep. The following day, the medium is changed to serum-free Neurobasal. On day 6–7, cells were transduced with the indicated lentivirus (1 $\times$ 10 e9/ml) at 1:1000 dilution overnight, and the medium was replaced the next day. After allowing an additional day of recovery, cells were pulsed for 30 min with 100 µM forskolin (Sigma # F6886) prior to fixation involving a 15 min pre-fix with 4% PFA in PBS added at a 1:1 ratio to the existing medium, followed by 10 min in 4% PFA solution only. Cells were then probed for p-CREB (i.e., cell signaling), BIII-tubulin (BioLegend MMS-435P) and the corresponding fluorescent secondary antibodies as indicated, as well as for DAPI. Cells were then imaged by fluorescent microscopy, and cell total corrected fluorescence of p-CREB in infected neurons was obtained using ImageJ. Statistical analyses were performed using Tukey's multiple comparisons test under one-way ANOVA (GraphPad Prism6 software). Multiplicity adjusted p-values are reported.

## Acknowledgements

We thank Dr Maryann Martone from the National Center for Microscopy and Imaging Research (NCMIR) for valuable discussions and critical comments on the manuscript. We thank MHE laboratory members at NCMIR for helpful discussion, especially Dr Tom Deerinck and Dr Angela Cone (NCMIR). We thank Hiro Hakozaki (NCMIR) for technical support. We thank Dr Harvey Karten, UCSD, for valuable discussions on the manuscript. We are particularly grateful to Dr Davide Dulcis for critical comments on the manuscript. We thank all members of the SST lab, especially: Dr Yuliang Ma for plasmids and valuable discussions, and Dr Kristofer J Haushalter for antibody-specificity immunoblots. We thank Dr Ayelet Gonen for helping with statistical analysis. This work was supported by National Institute of Health Grants DK054441 (to SST). This work was supported by grants from the NIH National Institute of General Medicine Sciences under award number P41GM103412

to Mark Ellisman, which funds the National Center for Microscopy and Imaging Research, and through award number 2 R01 GM082949 to Mark H Ellisman, which funds the Cell Image Library/ Cell Centered Database.

## Additional information

### Funding

| Funder | Grant reference number | Author |
|---|---|---|
| National Institute of Diabetes and Digestive and Kidney Diseases | DK054441 | Susan S Taylor |
| National Institute of General Medical Sciences | P41GM103412 | Mark H Ellisman |
| National Institute of General Medical Sciences | GM082949 | Mark H Ellisman |

The funders had no role in study design, data collection and interpretation, or the decision to submit the work for publication.

### Author contributions

RI, Conceptualization, Formal analysis, Supervision, Validation, Investigation, Visualization, Writing—original draft, Writing—review and editing; VL-R, EAB, TLS, Conceptualization, Formal analysis, Visualization; DF-M, Conceptualization, Formal analysis; CD, Formal analysis, Visualization; JLG, Conceptualization, Resources; MHE, Conceptualization, Resources, Funding acquisition, Visualization; SST, Conceptualization, Resources, Supervision, Funding acquisition, Investigation, Writing—review and editing

### Author ORCIDs

Eric A Bushong, http://orcid.org/0000-0001-6195-2433
Susan S Taylor, http://orcid.org/0000-0002-7702-6108

### Ethics

Animal experimentation: All experiments involving vertebrate animals conform to the National Institute of Health Guide for the Care and Use of Laboratory Animals (NIH publication 865-23, Bethesda, MD, USA) and were approved by the Institutional Animal Care and Use Committee (IACUC) of the University of California San Diego. Approved Animal Protocol Numbers: S03172m, S03182R.

## Additional files

### Major datasets

The following datasets were generated:

| Author(s) | Year | Dataset title | Dataset URL | Database, license, and accessibility information |
|---|---|---|---|---|
| Ronit Ilouz, Varda Lev-Ram, Eric A Bushong, Travis L Stiles, Dinorah Friedmann-Morvinski, Christopher Douglas, Geoffrey Goldberg, Mark H Ellisman, Susan S Taylor | 2016 | Data from: Isoform-specific subcellular localization and function of protein kinase A identified by mosaic imaging of mouse brain | http://www.cellimagelibrary.org/images/49451 | Publicly available at the Cell Centered Database (CIL: 49451) |
| Ilouz R, Lev-Ram V, Bushong EA, Stiles T, Friedmann-Mor- | 2016 | Data from: Isoform-specific subcellular localization and function of protein kinase A | http://www.cellimagelibrary.org/images/49651 | Publicly available at the Cell Centered Database (CIL: |

| | | | | |
|---|---|---|---|---|
| vinski D, Douglas C, Goldberg G, Ellisman MH, Taylor SS | | identified by mosaic imaging of mouse brain | | 49651) |
| Ilouz R, Lev-Ram V, Bushong EA, Stiles T, Friedmann-Morvinski D, Douglas C, Goldberg G, Ellisman MH, Taylor SS | 2016 | Data from: Isoform-specific subcellular localization and function of protein kinase A identified by mosaic imaging of mouse brain | http://www.cellimagelibrary.org/images/49453 | Publicly available at the Cell Centered Database (CIL:49453) |

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
