## [Decision Letter]

[Editors’ note: this article was originally rejected after discussions between the reviewers, but the authors were invited to resubmit after an appeal against the decision.]

Thank you for submitting your work entitled "Isoform-specific subcellular localizations and functions of protein kinase A identified by mosaic brain mapping" for consideration by *eLife*. Your article has been favorably evaluated by Eve Marder (Senior Editor) and three reviewers, one of whom is a member of our Board of Reviewing Editors. The following individual involved in review of your submission has agreed to reveal their identity: Ted Abel (Reviewer #3).

Our decision has been reached after consultation between the reviewers. Based on these discussions and the individual reviews below, we regret to inform you that your work will not be considered further for publication in *eLife*.

The reviewers felt that the manuscript contains data that is potentially significant and important, but the joining of two separate studies, one a survey of localizations of RIbeta and RIIbeta throughout the brain, and the second a set of biochemical studies indicating the importance of subcellular location of the two subunits, ultimately does not work well, as indicated below. Neither study is as complete or adequately controlled as would be appropriate for *eLife*. The authors are encouraged to strengthen both studies and submit them separately to more specialized journals.

The manuscript by Ilouz et al. first introduces detailed maps of the regional, cellular, and subcellular locations of the RIbeta and RIIbeta regulatory subunits of the cAMP-dependent protein kinase. These two isoforms are largely brain-specific, and have been shown to impair aspects of brain function when they are deleted or down-regulated. The localization maps permit comparison of the differential localization of each of these two isoforms in cortex, hippocampus, striatum, and lower brain regions. The images have been loaded into the Cell-centered Data Base maintained by the National Center for Microscopy and Imaging Research.

The investigators found that RIbeta, but not RIIbeta localizes to the nucleus of a subset of neurons in different brain regions. The nuclear localization was confirmed by mini-SOG conversion and electron microscopy. Identifying broad and subcellular expression is something that's been overlooked for various components of the cAMP pathway for far too long, and this is a welcome and necessary study. The main findings are that RIΒ is more highly expressed in the dendrites and soma (around the nucleus), and that RIIB is mainly in the axons.

However, there are several deficiencies in the presentation:

1) The use of vague, inappropriate terminology and/or jargon is frequent and interferes with clear presentation of the results. For example:

a) In the Abstract: "[…]confirms striking differences between RIβ and RIIβ that we functionally validate." The concept of "functionally validating" "striking differences" is vague. I believe the authors mean that they have shown that different localization of the subunits can lead to differences in their functions. The authors should delete the phrase "that we functionally validate" from this sentence. The phrase in the later sentence; "To functionally validate nuclear localization[…]" should be changed to; "To show the functional significance of nuclear localization, we demonstrate that[…]"

b) The localization studies as presented are poorly controlled. Images of staining with secondary antibody only should be shown for the images in Figure 1, and for subsequent higher power images. Figure 1—figure supplement 2 should be added to Figure 1 proper, along with secondary only controls. The superposition of red and green makes the differential localization more obvious.

c) One reviewer felt that the control experiments performed on tissue culture cells transfected with the proteins of interest are inadequate. This reviewer felt that additional western blot controls showing the specificity of each antisera against brain extracts and secondary antibody staining of brain tissue sections should be included.

d) A complete survey of the distribution of RIβ and RIIβ in mouse brain should present staining patterns for both proteins in coronal and sagittal tissue sections to reveal brain areas not contained in the coronal sections of forebrain.

e) One reviewer felt that it is difficult to follow the text describing the data in Figure 3. The authors' conclusions would be strengthened if they used neuronal markers for dendrites and axons. Simple counterstaining with these markers would be an important addition to the study.

f) The data in Figure 4 argue that RIβ is expressed at higher levels than RIIβ in regions of the cerebellum. However, the immunofluorescence data presented in the study are not adequate by themselves to justify this conclusion. The impression of higher levels could be an artifact if the affinity of the RIβ antibodies is much higher than those for RIIβ. Independent verification of protein levels in these tissues should be added to the study.

g) Because of the light staining in higher resolution images in Figure 6, it would be especially important to include control images of staining with secondary alone.

h) The authors make the broad claim that RIβ is expressed in dendrites and the nucleus, while RIIβ is expressed in axons; however, they make this claim based upon a small number of brain regions and cell types. In particular, it appears that most of the regions they looked at (the cerebellum and hippocampus) are regions where RIβ was very highly expressed and RIIB was much lower. This raises the question of what happens when you look at an area where RIIβ is more highly expressed, such as the caudate and putamen. Does higher expression of RIIβ change its subcellular compartmentalization?

i) If RIIβ is more highly expressed in axons, why is there such strong expression in the soma in Figure 5? This seems to contradict the results.

j) The miniSOG experiments in Figure 7 are preliminary. Additional data showing the signals with secondary antibodies alone should be included. The results need to be quantified. How many nuclei/mitochondria were examined, and how many of those were positive for RIbeta?

The reviewers found several problems with the second part of the manuscript:

a) Beginning with the section of the Results “RIβ, but not RIIβ, controls PKA-dependent CREB phosphorylation that is required for Late-Long Term Potentiation (L-LTP)”, the writing becomes unclear and often laced with jargon.

a1) "[…]down-regulation of RIβ is associated with memory deficit." This statement is not accompanied by citation and needs more explanation.

a2) "[…]stimulated with forskolin, an adenylyl cyclase activator, which has been previously used as a culture model for L-LTP.." Despite the two citations, this treatment is not widely accepted as a "culture model" for L-LTP. It would be better for the authors not to try to conflate CREB phosphorylation with L-LTP because the two are not identical. There are many additional steps and processes involved in L-LTP, depending on how it is induced. CREB phosphorylation plays a role in any process in which induction of the CRE promoter is involved. It would be better to simply list some of those processes. The general importance of CREB phosphorylation for brain regulation is not disputed.

a3) "[…]These results support previous studies indicating the role of RIβ in L-LTP, and provide an insight into the cause of long-term memory deficit in R (AB) transgenic animal (Abel et al., 1997)." This statement needs to be expanded into a short paragraph with an explanation of the relationship between results with R(AB) animals and the experiment of the authors. *eLife* is a journal read by a diverse readership. Most readers will be unfamiliar with R(AB) transgenic animals. Similarly, the discussion of these mutants in the "Discussion" section is inadequate.

a4) In the subsection “RIβ mutation prevents RIβ -homodimer formation and perturbs AKAP binding” and in the Methods section. The description of the experiments with the Rluc assay is full of jargon and very hard to parse. The term PPI for "protein-protein interaction" should be discarded. Instead, a plain English description of each interaction measured should be given, for each phase of the experiment.

a5) In the subsection “RIβ mutation prevents RIβ -homodimer formation and perturbs AKAP binding”: "This complex formation could be dissociated upon forskolin treatment". This is sloppy English and can be confusing. The authors don't mean that "complex formation" could be dissociated. They mean that the complexes could be dissociated by forskolin treatment. This section needs thorough editing by a native English speaker.

a6) In the subsection “RIβ mutation prevents RIβ -homodimer formation and perturbs AKAP binding”, the authors need to clearly state that the same region of RIbeta that is altered by the L50R mutation is involved in interaction with every AKAP. Otherwise, the statement; "[…]support the notion that AKAP binding to RIβ-L50R is affected in general" is not clear.

b) The Discussion needs thorough editing.

b1) "Using primary neuronal cells, we show that down-regulation of the nuclear RIβ, but not RIIβ, decreased L-LTP related signaling as reported by CREB phosphorylation." CREB phosphorylation is not considered a reporter for L-LTP. It is involved in many other processes. Thus, this statement is seriously misleading.

b2) "Dissociated cultured cells often do not provide an accurate understanding of spatial localization." This is a misleading overstatement. The word "often" should be changed to "sometimes". Dissociated cultured cells have provided a great deal of information about subcellular localization of many proteins, that has been thoroughly validated in brain tissue.

b3) In the third paragraph of the Discussion, the authors present a rather rambling discussion of the specificity and roles of AKAPs, which is ultimately confusing. This part of the Discussion needs reorganization, starting with a description of the association of RIbeta with "dual-specificity" AKAPs and with those that are specific to RIbeta (if there are any). Are Type I AKAPs specific for association with RI?

c) Figure 8—figure supplement 1 and 2 should be added to Figure 8 proper. Both the validation of the shRNAs and the quantification of fluorescence are integral to the experiment.

d) The resolution of the images in Figure 8 appear low and should be augmented with biochemical data (Western blots) supporting the changes in phosphorylation of CREB. It would be informative to know what happens to phosphorylation of Ser 142 on CREB. This is a Cam K II site and should serve as a good negative control

The comments above are a synthesis of the three reviews. The comments of Ted Abel are presented in toto below:

*Reviewer #3:*

Summary. In this manuscript, Ilouz and colleagues seek to characterize and differentiate the expression patterns of PKA regulatory subunits RIβ and RIIβ through high resolution confocal mosaic brain mapping and electron microscopy. The idea is simple and straightforward, to identify differences in broad expression and subcellular expression through multiple imaging modalities, and then begin to probe to see if these differences in expression can result in unique downstream changes in biochemistry and/or behavior. Identifying broad and subcellular expression is something that's been overlooked for various components of the cAMP pathway for far too long, and this is a welcome and necessary study. The main findings are that RIβ is more highly expressed in the dendrites and soma (around the nucleus), and that RIIβ is mainly in the axons. They then go to show that only down regulation of RIβ changes pCREB signaling while RIIβ does not, further confirming the differences in compartmentalization between these two subunits. Interestingly, they link this difference in expression to a recently discovered familial disease that results from a mutation in the RIβ subunit which provides further evidence demonstrating how important this compartmentalization is.

1) The authors make the broad claim that RIβ is expressed in dendrites and the nucleus, while RIIβ is expressed in axons; however, they make this claim off looking at a small number of brain regions and cell types. In particular, it appears that most of these regions they looked at (the cerebellum and hippocampus) are regions where RIβ were very highly expressed and RIIβ was much lower. My question is what happens when you look at an area where RIIβ is more highly expressed, such as the caudate and putamen. Does higher expression of RIIβ change its subcellular compartmentalization?

2) If RIIβ is more highly expressed in axons, why is there such strong expression in the soma in Figure 5? This seems to contradict the results.

[Editors’ note: what now follows is the decision letter after the authors submitted for further consideration.]

Thank you for resubmitting your work entitled "Isoform-specific subcellular localizations and functions of protein kinase A identified by mosaic brain mapping" for further consideration at *eLife*. Your article has been favorably evaluated by Eve Marder (Senior Editor) and three reviewers, one of whom is a member of our Board of Reviewing Editors.

The manuscript has been improved but there are some remaining issues that need to be addressed before acceptance, as outlined below:

Summary:

Using isoform-specific antibodies against the two major regulatory subunits of cAMP-dependent protein kinase, RIβ and RIIβ, the authors generated high-resolution large-scale immunohistochemical mosaic brain images that provide global views of the distribution of the two isoforms in several brain regions, including the hippocampus and cerebellum. The isoforms concentrate in discrete brain regions and the results reveal distinct patterns of subcellular localization. RIβ is enriched in dendrites and colocalizes with MAP2, whereas RIIβ is concentrated in axons. Using correlated light and electron microscopy the authors confirm mitochondrial and nuclear localization of RIβ in cultured neurons. To show the functional significance of nuclear localization of RIβ, the authors demonstrate that down-regulation of RIβ, but not RIIβ, decreased CREB phosphorylation. The study reveals how PKA isoform specificity is defined by precise localization.

The authors have been responsive to the concerns of reviewers of previous versions of this manuscript about the immunohistochemical study by adding controls to address antibody specificity and improving the focus and clarity of the manuscript with changes in the text and the addition of several supplemental figures. This part of the manuscript is substantially improved. There are still some concerns about the functional study that can be addressed by appropriate revisions.

Essential revisions:

1) The description of the knockdown experiment shown in Figure 8 is not yet adequate. The efficiency of the shRNAs was tested by co-transfection with "PRKAR2beta" or "PRKAR1beta" and infection with the shRNA lentiviruses. However, very little detail is given for how this experiment was carried out either in the Results section or in the methods. Figure 8 appears to show that only one of 3 shRNAs were effective against R2beta when tested by cotransfection in a cultured cell line. One has to assume that only shRNA#2 was used in the experiments in neurons shown in Figure 8; however, this is not explicitly stated.

2) More troubling is that the authors have not reported any test to check whether the appropriate R subunits are, in fact, knocked down in the neurons after infection with the shRNAs. The authors should be able to carry out infections and then stain the infected cultures for RIβ or RIIβ and compare the intensity of staining between non-infected and infected neurons. Without information about the actual extent of knockdown of each subunit in the infected neurons, it is not possible to know how significant the difference is between p-CREB after infection with RIβ vs. RIIβ shRNAs. It is important to know how effective the knockdown of each subunit was in the neurons, in order to interpret the changes in p-CREB appropriately.

---

## [Author Response]

[Editors’ note: the author responses to the first round of peer review follow.]

*The reviewers felt that the manuscript contains data that is potentially significant and important, but the joining of two separate studies, one a survey of localizations of RIbeta and RIIbeta throughout the brain, and the second a set of biochemical studies indicating the importance of subcellular location of the two subunits, ultimately does not work well, as indicated below. Neither study is as complete or adequately controlled as would be appropriate for eLife. The authors are encouraged to strengthen both studies and submit them separately to more specialized journals.*

We appreciate the editor’s and the reviewers’ comments regarding this part of the manuscript. After re-reading the manuscript along with all your helpful comments, as well as discussing this part with one of the identified reviewers, Ted Abel, we agree that it would be best to remove the last experiment (Figure 9) that relates to the RIβ L50R mutation, found in patients with a neurodegenerative disease. This additional data seems to deflect from the message of the manuscript and provides an impression that two separate studies are presented in this manuscript. As the paper is now structured we present the regional, cellular and subcellular spatial localization of RIβ and RIIβ that is provided by the high-resolution large-scale mosaic images using isoform-specific antibodies. We then present a few specific and significant examples of localization and functional consequences that emerged from the large data set. One of the examples that we focused on was the nuclear localization of RIβ, the least studied PKA isoform, in different brain regions. We further investigated the subcellular localization of RIβ using miniSOG labeling that allowed for correlating light and electron microscopy. We find RIβ at the mitochondria, as we reported earlier, as well as at the nucleus, modifying the existing dogma regarding PKA isoforms in the nucleus. The images suggested a potential functional distinction between the two isoforms in the nucleus. To support this we showed that down-regulation of RIβ, but not RIIβ, in hippocampal cultures decreased pCREB, which we use as an example of a nuclear PKA substrate. Taken together, we present a multi-scale approach, going from the global views of whole brain sections, which allow identification of subcellular features, without losing the context of the brain regions, to the validation of the subcellular localization using electron microscopy and correlation of this localization with functional differences of PKA isoforms. Obviously, there are many other ways in which this data, that defines comprehensively the unique enrichment of RIβ vs. RIIβ across multiple brain regions, can be utilized to drive new hypotheses. We feel that by ending the manuscript here as well as addressing the reviewers’ comments on the functional data (as provided by point-by-point response below) the focus and clarity of the manuscript has significantly improved.

*The manuscript by Ilouz et al. first introduces detailed maps of the regional, cellular, and subcellular locations of the RIbeta and RIIbeta regulatory subunits of the cAMP-dependent protein kinase. These two isoforms are largely brain-specific, and have been shown to impair aspects of brain function when they are deleted or down-regulated. The localization maps permit comparison of the differential localization of each of these two isoforms in cortex, hippocampus, striatum, and lower brain regions. The images have been loaded into the Cell-centered Data Base maintained by the National Center for Microscopy and Imaging Research.*

The investigators found that RIbeta, but not RIIbeta localizes to the nucleus of a subset of neurons in different brain regions. The nuclear localization was confirmed by mini-SOG conversion and electron microscopy. Identifying broad and subcellular expression is something that's been overlooked for various components of the cAMP pathway for far too long, and this is a welcome and necessary study. The main findings are that RIΒ is more highly expressed in the dendrites and soma (around the nucleus), and that RIIB is mainly in the axons.

*However, there are several deficiencies in the presentation:*

[…] 1) The use of vague, inappropriate terminology and/or jargon is frequent and interferes with clear presentation of the results. For example:

*a) In the Abstract: "[…]confirms striking differences between RIβ and RIIβ that we functionally validate." The concept of "functionally validating" "striking differences" is vague. I believe the authors mean that they have shown that different localization of the subunits can lead to differences in their functions. The authors should delete the phrase "that we functionally validate" from this sentence. The phrase in the later sentence; "To functionally validate nuclear localization[…]" should be changed to; "To show the functional significance of nuclear localization, we demonstrate that[…]"*

We deleted the phrases: “that we functionally validated” and “striking differences” from the Abstract. We also changed the phrase: “To functionally validate nuclear localization” to “To show functional significance of nuclear localization, we demonstrate that[…]” as the reviewer suggested. We thank the reviewer for helping to clarify the content of the Abstract. We also took out the LTP related signaling from the Abstract, in response to the reviewer’s later comment. We now mention CREB phosphorylation in correlation with L-LTP related signaling only in the Discussion.

*b) The localization studies as presented are poorly controlled. Images of staining with secondary antibody only should be shown for the images in Figure 1, and for subsequent higher power images. Figure 1—figure supplement 2 should be added to Figure 1 proper, along with secondary only controls. The superposition of red and green makes the differential localization more obvious.*

*c) One reviewer felt that the control experiments performed on tissue culture cells transfected with the proteins of interest are inadequate. This reviewer felt that additional western blot controls showing the specificity of each antisera against brain extracts and secondary antibody staining of brain tissue sections should be included.*

Antibody specificity:

Determining antibody specificity and selectivity are indeed the most important requirements for any immunohistochemical study. We performed a comprehensive analysis with all of these controls to determine specificity and selectivity of the primary and secondary antibodies used in this study. Only after we were convinced that the antibodies are specific and selective did we carry out the large-scale high-resolution immunohistochemical imaging. We agree that the original manuscript lacks the negative controls for antibody validation but they were all done routinely for every experiment. We appreciate the reviewer’s concerns, and the requested controls are now included or specifically mentioned in our revised manuscript. These controls for antibody specificity and selectivity are now discussed comprehensively and up front at the beginning of the manuscript (Results section:” Overview of the regional distribution of RIβ and RIIβ across brain regions”) and the control images are included in Figure 1—figure supplement 1. Negative controls for the tissue staining were also added in a new figure. Figure 1—figure supplement 2.

To address the reviewer’s concerns regarding antibody specificity:

1) Controls for secondary antibodies alone were done for each secondary antibody used in this study. The secondary antibodies were specific and detected nothing as proven by the black images. This negative control was done on tissue sections as well as on cultured cells.

The primary antibodies used in this study are commercially available and western blot analyses are shown in the datasheet of the companies. We observed a single band at the known molecular weight of the PKA regulatory subunit when the proteins are denatured. Our goal was to use these antibodies for IHC, where the antigens are at their native conformation, and therefore we performed additional experiments using dot blots of purified proteins to validate specificity and selectivity of the primary antibodies for proteins in their native conformation. We overexpressed RIβ-MKO2 or RIα-MKO2 and showed that the RIβ antibody detects only the RIβ overexpressed protein. The RIβ antibody staining was also overlapping with the MKO2 signal. To further validate that the RIβ antibody does not cross-react with its highly homologous regulatory subunit, RIα, we overexpressed RIα-MKO2 and showed that it is detected by the RIα antibody but not by the RIβ antibody even when the laser power was significantly increased (This experiment is shown is Figure 1—figure supplement 1). Control of secondary antibody only was done along with the experiment using similar laser power and higher. The secondary antibody did not detect anything and the images were black. The same experiment was done to validate the RIIβ antibody specificity.

An additional negative control that was not mentioned in the original manuscript, was an experiment where we incubated the RIβ antibody with purified full length human RIβ protein, that we purify routinely in our lab, to block the antibody. The pre-adsorption antibody resulted in loss of the staining on the tissue. The images were black. This experiment demonstrates that the antibody is specific for the immunogen. The same control was done for RIIβ antibody. This control is also now included in Figure 1—figure supplement 1.

We did not previously include checks for cross reactivity between RI and RII antibody because only α and β forms of either RI or RII are highly homologous. This control is now included in Figure 1—figure supplement 1, where the dot blot shows that there is not cross reactivity between RI and RII subunits.

We find that the primary and secondary antibodies used in our study are reproducible as we tested different lot numbers. Reproducibility is an important criterion for validation and standardization. This note is important as we expect that this paper will be a good reference for these specific antibodies and is included now in the manuscript.

Figure 1—figure supplement 2 is added now to Figure 1. As the reviewer commented adding this supplement figure to the main figure makes the differential localization more obvious.

*d) A complete survey of the distribution of RIβ and RI β in mouse brain should present staining patterns for both proteins in coronal and sagittal tissue sections to reveal brain areas not contained in the coronal sections of forebrain.*

It was not our intention to provide an atlas for the protein distribution of these PKA isoforms. To clarify this we also changed the title and replaced mapping with imaging. For these initial studies we focused primarily on brain regions where we anticipated that these PKA isoforms would be highly concentrated and that also correlate with the isoform specific functions that were defined by the RIβ or RIIβ knockout studies. Our major focus, therefore, was to provide cellular and subcellular analyses of protein distribution in the hippocampus and the cerebellum without losing the tissue context. We have clarified the reason we selected these brain regions in the Results section. These imaging maps are rich in information and we focused on only a few and specific examples to correlate the data with electron microscopy and functional studies. These comprehensive maps, uploaded online, can drive new hypotheses on many of the cellular and subcellular regions that are included in these maps but were not discussed in the current manuscript.

*e) One reviewer felt that it is difficult to follow the text describing the data in Figure 3. The authors' conclusions would be strengthened if they used neuronal markers for dendrites and axons. Simple counterstaining with these markers would be an important addition to the study.*

We agree that immunohistochemical high magnification images usually require co-staining with specific markers to validate dendrite or axon co-localization. The advantage of the high–resolution large-scale mosaic images is that these comprehensive maps allow us to analyze protein localization at a gross structural level and at the same time provide subcellular details, thereby allowing studying protein localization at different scales simultaneously. The images as simply shown in Figure 3 were taken from the full maps and, therefore, lack the complete picture. We deduced that RIIβ is in axons by combining insights from the low and high-resolution scales. When we trace a whole axonal pathway, we can also clearly see boutons, which are part of the axonal structure. In addition, this staining does not co-localize with the dendritic staining (for example: CA3 region). We have emphasized this now in the revised manuscript (Results section: RIIβ is concentrated in axons whereas RIβ is concentrated in dendrites and somata in various subfields of the hippocampus”.

*f) The data in Figure 4 argue that R β is expressed at higher levels than RIIβ in regions of the cerebellum. However, the immunofluorescence data presented in the study are not adequate by themselves to justify this conclusion. The impression of higher levels could be an artifact if the affinity of the RIβ antibodies is much higher than those for RIIβ. Independent verification of protein levels in these tissues should be added to the study.*

The conclusion that RIβ is the predominant regulatory subunit at the cerebellum was not made based on the immunofluorescence data presented in Figure 4 but on a western blot analysis showing the expression levels of all PKA regulatory subunits in the cerebellum (Weisenhaus, 2010). In addition, we also mentioned that in situ hybridization data showed much less hybridization of RIIβ compared to RIβ at the cerebellum (Cadd and McKnight, 1989). Our results are quite consistent with McKnight’s earlier analysis of protein expression in these regions. We revised the text discussing the cerebellum staining to prevent an impression that we concluded that RIβ is the predominant isoform at the cerebellum based on the immunostaining.

*g) Because of the light staining in higher resolution images in Figure 6, it would be especially important to include control images of staining with secondary alone.*

We have now included these controls in Figure 6—figure supplement 1. The nuclear localization of RIβ was also validated by electron microscopy (Figure 7).

*h) The authors make the broad claim that RIβ is expressed in dendrites and the nucleus, while RIIβ is expressed in axons; however, they make this claim based upon a small number of brain regions and cell types. In particular, it appears that most of the regions they looked at (the cerebellum and hippocampus) are regions where RIβ was very highly expressed and RIIβ was much lower. This raises the question of what happens when you look at an area where RIIβ is more highly expressed, such as the caudate and putamen. Does higher expression of RIIβ change its subcellular compartmentalization?*

This is a very important point. Because we did not do whole brain mapping we cannot answer this question comprehensively but these are clearly major questions that need to be explored in future studies. However, if we zoom in on the striatum where RIIβ is enriched we still see that it is excluded from nuclei whereas RIβ is still present in most nuclei. Nuclear localization thus does not appear to be influenced by the level of expression. We do not claim that RIβ, as opposed to RIIβ, is always enriched in dendrites of every neuron, but this pattern was consistently seen in all of the neurons that we examined here.

*i) If RIIβ is more highly expressed in axons, why is there such strong expression in the soma in Figure 5? This seems to contradict the results.*

A significant body of evidence now exists to suggest RNA translation occurs within axons. This is in contrast to the classical thought that the cell body is the exclusive source of axonal proteins. In any case, we believe that the localization of RIIβ in axons as well as the cell body is not contradictory; as it can be in both. We performed an additional experiment with different organelle markers and found that the strong labeling of RIIβ in the cell body comes from the Golgi apparatus.

*j) The miniSOG experiments in Figure 7 are preliminary. Additional data showing the signals with secondary antibodies alone should be included. The results need to be quantified. How many nuclei/mitochondria were examined, and how many of those were positive for RIbeta?*

There seems to be a misunderstanding about the miniSOG methodology, which we hope to clarify through our responses below and by revising the text in the manuscript. Contrary to what the reviewer suggested, no primary or secondary antibodies are involved in this method. The miniSOG (for mini Singlet oxygen generator) is a genetically encoded tag, designed and validated by Tsien and Ellisman, that allows us doing correlated light and electron microscopy. This is an essential feature of this innovative probe and provides a way for achieving higher resolution (Shu, X, pLOS biology, 2011). RIβ cDNA was fused to a construct containing the miniSOG tag. Many transfected neuronal cells and adjacent non- transfected neuronal cells were identified by fluorescent microscopy. The advantage of this probe is that the same region that was viewed by light microscopy can be analyzed with electron microscopy following photo-oxidation. In Figure 7 we provide EM images of 2 different transfected cells and controls for their adjacent non-transfected cells, however, 126 transfected darkened mitochondria and 71 non-transfected light mitochondria were counted. This method has been proven reliable and widely used, as the reference paper has been cited now over 100 times in PubMed.

*The reviewers found several problems with the second part of the manuscript:*

*a) Beginning with the section of the Results “RIβ, but not RIIβ, controls PKA-dependent CREB phosphorylation that is required for Late-Long Term Potentiation (L-LTP)”, the writing becomes unclear and often laced with jargon.*

We address some of the concerns below but most of the unclear text has now been removed as indicated above. We include the nuclear localization and the axon/dendrite localization as two observations that we believed were important to functionally confirm. We specifically address the concerns about CREB phosphorylation as a reporter for LTP. Here we agree with the reviewer and now simply use CREB phosphorylation as an example of a nuclear substrate for PKA.

*a1) "[…]down-regulation of RIβ is associated with memory deficit." This statement is not accompanied by citation and needs more explanation.*

This statement has now been removed as we emphasize better the rational of doing the CREB phosphorylation experiment. We removed the CREB phosphorylation as a reporter for L-LTP and the association of RIβ with memory deficit as a rational for the CREB phosphorylation experiment.

*a2) "[…]stimulated with forskolin, an adenylyl cyclase activator, which has been previously used as a culture model for L-LTP.." Despite the two citations, this treatment is not widely accepted as a "culture model" for L-LTP. It would be better for the authors not to try to conflate CREB phosphorylation with L-LTP because the two are not identical. There are many additional steps and processes involved in L-LTP, depending on how it is induced. CREB phosphorylation plays a role in any process in which induction of the CRE promoter is involved. It would be better to simply list some of those processes. The general importance of CREB phosphorylation for brain regulation is not disputed.*

We did not intend to conflate CREB phosphorylation with L-LTP. We agree with the reviewer that there are many additional steps and processes involved in L-LTP. We therefore re-wrote this section in the Results and specifically removed all reference to LTP. Instead we simply state that we wanted to investigate whether down-regulation of RIβ would result in impairment of CREB phosphorylation, a downstream PKA substrate at the nucleus.

*a3) "[…]These results support previous studies indicating the role of RIβ in L-LTP, and provide an insight into the cause of long-term memory deficit in R (AB) transgenic animal (Abel et al., 1997)." This statement needs to be expanded into a short paragraph with an explanation of the relationship between results with R(AB) animals and the experiment of the authors. eLife is a journal read by a diverse readership. Most readers will be unfamiliar with R(AB) transgenic animals. Similarly, the discussion of these mutants in the "Discussion" section is inadequate.*

As indicated earlier we have removed the link between L-LTP and CREB. We do not mention the experiments done with R(AB) transgenic animals.

*b) The Discussion needs thorough editing.*

*b1) "Using primary neuronal cells, we show that down-regulation of the nuclear RIβ, but not RIIβ, decreased L-LTP related signaling as reported by CREB phosphorylation." CREB phosphorylation is not considered a reporter for L-LTP. It is involved in many other processes. Thus, this statement is seriously misleading.*

We agree with the reviewer that CREB phosphorylation is involved in many other processes. We therefore re-wrote this section in the Discussion and specifically removed all reference to LTP. Instead we simply state that we wanted to investigate whether down-regulation of RIβ would result in impairment of CREB phosphorylation, a downstream PKA substrate at the nucleus.

*b2) "Dissociated cultured cells often do not provide an accurate understanding of spatial localization." This is a misleading overstatement. The word "often" should be changed to "sometimes". Dissociated cultured cells have provided a great deal of information about subcellular localization of many proteins, that has been thoroughly validated in brain tissue.*

This word has been changed. This a good point.

*b3) In the third paragraph of the Discussion, the authors present a rather rambling discussion of the specificity and roles of AKAPs, which is ultimately confusing. This part of the Discussion needs reorganization, starting with a description of the association of RIbeta with "dual-specificity" AKAPs and with those that are specific to RIbeta (if there are any). Are Type I AKAPs specific for association with RI?*

Thank you for this comment. We have clarified the discussion about the role of AKAPs and re-wrote the fourth paragraph of the Discussion. Type I AKAPs are specific for RI subunits.

*c) Figure 8—figure supplement 1 and 2 should be added to Figure 8 proper. Both the validation of the shRNAs and the quantification of fluorescence are integral to the experiment.*

We agree with this comment. The supplementary figures have been included in the main Figure 8.

*d) The resolution of the images in Figure 8 appear low and should be augmented with biochemical data (Western blots) supporting the changes in phosphorylation of CREB. It would be informative to know what happens to phosphorylation of Ser 142 on CREB. This is a Cam K II site and should serve as a good negative control*

This experiment has been carefully controlled and quantified. We have now emphasized it in the manuscript. We could do a western blot, however, it would not be as accurate as quantifying the intensity of CREB phosphorylation in infected neurons. Due to the robust survival capacity of non-neuronal cells, trace gila present in initial preparations were an unavoidable component of these cultures. Non-infected cells remain as well. To specifically assay CREB phosphorylation in infected cells, the signal was quantified only in cells positively stained for βIII-tubulin (neuronal marker, red) that expressed GFP (marker for lentivirus infection). CREB phosphorylation was quantified in the specified cells as a measure of pCREB signal intensity per the area of the nucleus as delineated by DAPI. We have internal controls in each experiment comparing CREB phosphorylation on Ser133 of infected neurons to non-infected neurons. We show statistically significant data of infected neurons with shRNA against RIβ compared to the scrabbled shRNA (p<0.0001). We show statistically significant data of neurons infected with shRNA against RIβ compared to RIIβ (p<0.001). 50 neurons were counted in each experiment. Since we have internal controls as well as two negative controls (scrabbled and RIIβ) and the data is statistically significant we are confident that the data is properly controlled.

[Editors’ note: the author responses to the re-review follow.]

*Essential revisions:*

*1) The description of the knockdown experiment shown in Figure 8 is not yet adequate. The efficiency of the shRNAs was tested by co-transfection with "PRKAR2beta" or "PRKAR1beta" and infection with the shRNA lentiviruses. However, very little detail is given for how this experiment was carried out either in the Results section or in the methods. Figure 8 appears to show that only one of 3 shRNAs were effective against R2beta when tested by cotransfection in a cultured cell line. One has to assume that only shRNA#2 was used in the experiments in neurons shown in Figure 8; however, this is not explicitly stated.*

We revised the sections describing this experiment to provide detailed information about the experiment and a more adequate description of results.

1) We edited the Materials and methods paragraph. This now includes:

“293T cells were co-transfected with either PRKARIβ or PRKARIIβ rat cDNA (Origene) and with scrambled shRNA (Σ) or shRNAs targeting PRKARIβ (5’- CGGCAGAAGTCAAACTCACAGTGTGATTC-3’, purchased from Origene) or PRKARIIβ (#1-5’-CGTCATCGACAGAGGAACATT-3’; #2-5’-CGATGCAGAGTCCAGGATAAT-3’ and # 3-5’-CCTTCAGGAGAATAATAGTAA-3’). […] Lentiviral particles were then prepared as previously described (Ikawa et al., 2003).”

2) We edited the Results section that describes this experiment:

“We first confirmed the knockdown efficiency by co-transfecting 293T cells with cDNA encoding either RIβ or RIIβ genes, together with a scrambled shRNA (shRNA IRR) as a negative control or shRNAs targeting either RIβ or RIIβ. Western blot analysis confirmed knockdown efficiency of RIβ at the protein level (Figure 8—figure supplement 1). Western blot analysis indicated that an efficient RIIβ knockdown had been achieved using shRNA RIIβ clone #2 (Figure 8—figure supplement 1). This clone was used in the subsequent experiments to knockdown RIIβ.”

3) We also edited the figure legend to Figure 8—figure supplement 1 describing this experiment.

We have also moved this experiment to a new figure. It is now in Figure 8—figure supplement 1 figure that provides a comprehensive knockdown efficiency validation as described in our response to comment #2 by the reviewer.

*2) More troubling is that the authors have not reported any test to check whether the appropriate R subunits are, in fact, knocked down in the neurons after infection with the shRNAs. The authors should be able to carry out infections and then stain the infected cultures for RIβ or RIIβ and compare the intensity of staining between non-infected and infected neurons. Without information about the actual extent of knockdown of each subunit in the infected neurons, it is not possible to know how significant the difference is between p-CREB after infection with RIβ vs. RIIβ shRNAs. It is important to know how effective the knockdown of each subunit was in the neurons, in order to interpret the changes in p-CREB appropriately.*

We agree with the reviewers and with the suggested experiment and have included a new supplementary figure: Figure 8—figure supplement 1. The figure provides a comprehensive validation for shRNA knockdown efficiency and includes:

A. and B. We first confirmed knockdown efficiency by co-transfecting 293T cells with cDNA encoding either RIβ (A) or RIIβ (B) genes, together with a scrambled shRNA as a negative control or shRNAs targeting either RIβ or RIIβ. Western blot analysis confirmed knockdown efficiency of shRNAs against RIβ and RIIβ at the protein level. The two western blots were part of Figure 8 in the original manuscript. We decided to include this information as part of a comprehensive figure showing shRNA knockdown efficiency.

C. The reviewer’s suggested experiment is now included. To validate the knockdown efficiency of shRNA against RIβ or RIIβ in primary cultured neurons we infected cells with shRNAs targeting either RIβ or RIIβ and used scrambled shRNA as a negative control. Cells were immunostained with either RIβ or RIIβ specific antibodies. Infected cells expressing shRNAs were identified by the presence of GFP. Cells expressing shRNA against RIβ or RIIβ display a significant reduction in the cell fluorescence intensity produced by their specific antibodies as compared to cells infected with a negative control. The results indicate a significant inhibition of RIβ or RIIβ expression in cells infected with shRNA against a corresponding gene. Representative cells are shown.

D. Quantification analyses to validate knockdown efficiency are also now provided. Knockdown efficiency was determined by measuring cell fluorescence intensity of the signal from RIβ or RIIβ specific antibodies. The immunofluorescence signal was reduced by shRNA against RIβ or RIIβ as compared to cells infected with a negative control. GFP fluorescence intensity was used to normalize the quantification analyses. Corrected total cell fluorescence was measured for each condition using imageJ. The shRNAs against RIβ or RIIβ demonstrate 70% or 80% knockdown efficiency, respectively in neuronal cells. P-value <0.0001, Student’s t-test.

We have revised the section in the Results (subsection “RIβ, but not RIIβ, contributes PKA-dependent CREB phosphorylation”) and in the Materials and methods (subsection “Image deposition into the cell centered database”) and the Figure 8—figure supplement 1 legend to provide a detailed description about this figure.